# Modern slavery and the race to fish

David Tickler [1,2], Jessica J. Meeuwig[1], Katharine Bryant[2], Fiona David[2], John A.H. Forrest[1,2], Elise Gordon[2], Jacqueline Joudo Larsen[2], Beverly Oh[1,2], Daniel Pauly[3], Ussif R. Sumaila[4] & Dirk Zeller[5]

Marine fisheries are in crisis, requiring twice the fishing effort of the 1950s to catch the same quantity of fish, and with many fleets operating beyond economic or ecological sustainability. A possible consequence of diminishing returns in this race to fish is serious labour abuses, including modern slavery, which exploit vulnerable workers to reduce costs. Here, we use the Global Slavery Index (GSI), a national-level indicator, as a proxy for modern slavery and labour abuses in fisheries. GSI estimates and fisheries governance are correlated at the national level among the major fishing countries. Furthermore, countries having documented labour abuses at sea share key features, including higher levels of subsidised distant-water fishing and poor catch reporting. Further research into modern slavery in the fisheries sector is needed to better understand how the issue relates to overfishing and fisheries policy, as well as measures to reduce risk in these labour markets.

[1] Marine Futures Lab, School of Biological Sciences, University of Western Australia, Crawley, WA 6009, Australia. [2] Walk Free Foundation, Perth, WA 6009, Australia. [3] Sea Around Us, Institute for the Oceans and Fisheries, University of British Columbia, Vancouver, BC V6T 1Z4, Canada. [4] Fisheries Economics Research Unit, Institute for the Oceans and Fisheries, University of British Columbia, Vancouver, BC V6T 1Z4, Canada. [5] Sea Around Us – Indian Ocean, School of Biological Sciences, University of Western Australia, Crawley, WA 6009, Australia. Correspondence and requests for materials should be addressed to D.T. (email: david.tickler@research.uwa.edu.au)

Since the mid-1990s, global marine fisheries catches have steadily decreased[1] while fishing effort has continued to increase, leading to intense competition, declining catch-per-unit-of-effort and fisheries profitability, and the over-exploitation of many stocks[1–4]. The consequent race to fish has been exacerbated by harmful government subsidies that enable fishing effort to persist beyond bio-economic limits[5]. The underlying pattern of decline has been masked in the officially reported data by inconsistent data reporting from some areas of the world[1,6,7] and by a presentist bias[8] that assumes improved catch reporting equals increased catches[9]. The resultant overly optimistic trend in government data has fostered suboptimal policies, in particular the allocation of resources to harmful capacity-enhancing subsidies rather than enforcement or stock rebuilding[1,6,7]. Failure to manage fisheries sustainably has serious implications for human welfare, as fish (here meaning finfish and invertebrates) provide billions of people with protein and vital nutrients[10], as well as employment and livelihoods for hundreds of millions of people[11].

Falling productivity and financial returns in commercial fisheries can pressure vessels to cut operating costs, at the extreme by fishing illegally, circumventing licensing costs and catch limits[12], and by reducing expenditure on crew pay, safety and living conditions. Estimates of fishing labour costs suggest that they comprise 30–50% of total fishing costs[4,13]. The large contribution of labour to fishing costs suggests that, in addition to government subsidies received for fuel, vessel operators can capture a significant additional subsidy by aggressively reducing expenditure on crew, for example, by non-compliance with labour and safety standards or by withholding pay.

The push to reduce operating expenses to maintain profitability has occurred in the context of rising living standards and employment expectations in industrialised fishing countries, leading to domestic crew shortages and higher wage demands[14,15]. Concurrently, the political marginalisation of coastal, small-scale fisheries throughout the developing world[16], exacerbated by population growth, has contributed to a surplus of domestic and migrant labour in developing countries[17–19]. This has polarised labour supply and demand between developed/emerging and developing economies, forcing people in the latter group to engage in any work available, including as fishing crew in an industry highly motivated to cut costs and that often operates out of reach of enforcement agencies[14,20].

Given the nature of working at sea, labour conditions of fishing crews are difficult to monitor. Supported by reefers and supply ships, fishing vessels can remain at sea for months during which time the crew may be unable to disembark[21], with living and working conditions on such vessels generally beyond the oversight of regulators[15]. Given jurisdictional complexities, it is also often unclear in which country a crew member can seek redress in cases of abuse[22]. While flag-state responsibility matters, the growing use of flags of convenience further weakens the capacity to enforce regulations[23,24]. These factors facilitate the use of exploitative employment practices to reduce labour costs at the expense of worker pay, safety and freedom[25].

The isolation of workers at sea makes the extent of labour issues in fisheries difficult to quantify. In recent years, however, high profile media investigations have identified a number of cases of extreme labour abuses in fisheries, some involving hundreds of fishing crew. Investigations of the Thai, Taiwanese and South Korean fishing industries identified cases of human trafficking, forced confinement, physical abuse and even murder[26–30]. These incidents have not been confined just to the high seas or the waters of weaker jurisdictions. Some of the cases involving South Korean vessels took place while under charter in New Zealand waters[31–33]. There have also been allegations of human trafficking and debt bondage of African and Asian crew on domestic vessels in British and Irish fisheries[34–36] and trafficking and confinement among South East Asian fishers employed in US fisheries in Hawaii[37]. The US State Department lists 40 countries as source, destination or transit countries for human trafficking in fisheries[38], and vessels exploiting fishing crew have been encountered in the waters of Indonesia, Papua New Guinea, Russia and South Africa, as well as New Zealand[25,39–41]. Labour rights abuses in fisheries appear widespread and serious, in many cases meeting the definition of modern slavery.

Modern slavery is defined, for the purposes of measurement, by the International Labour Organisation and the Walk Free Foundation (WFF) as "any situation of exploitation that a person cannot refuse or leave because of threats, violence, coercion, deception, and/or abuse of power". This includes "forced labour, debt bondage, forced marriage, slavery and slavery-like practices and human trafficking"[42]. As the United Nations Office on Drugs and Crime notes, "the common denominator of these crimes is that they are all forms of exploitation in which one person is under the control of another"[43]. At present, at least 40 million people are estimated to be trapped in modern slavery in textile, agriculture, construction and fisheries sectors, as well as in the sex industry and in forced marriage[42]. Modern slavery exists at the extreme end of a spectrum of exploitative and abusive labour practices, many of which remain legal in the jurisdiction in which they occur and/or are entered into voluntarily by workers[14,20,25,44]. Commentators rightly argue that a narrow focus on slavery, without broader attention to the needs, ambitions and vulnerability of workers, risks inadequate or even counter-productive responses[25,44]. However, unlike other labour issues, slavery is universally illegal, with prohibitions enshrined in global agreements including the 1926 Slavery Convention and the United Nations Convention on the Law of the Sea. It can therefore, in principal, be addressed using existing legal frameworks and instruments, and measures that identify and tackle slavery may have a positive impact on other less explicit forms of labour abuse. Importantly for the purpose of identifying global patterns, the above definition of modern slavery has allowed country-level estimates of the prevalence of modern slavery to be made by the Global Slavery Index (GSI)[45]. While not directly quantifying slavery at sea, the GSI data provide a proxy for analysing the relationship between the prevalence of slavery-like practices in a country and fisheries' characteristics at the global level, which may help identify drivers and policy priorities.

In addition to the structural elements in industrial fisheries that may incentivise and enable modern slavery and labour rights abuses, the global seafood trade is another critical dimension of the issue. Seafood is the world's most widely traded food commodity[46], involving complex supply chains, with the chain of custody often passing through several intermediaries and countries before reaching the consumer. Traceability issues often arise before the fish even enter the supply chain, with the widely used practice of transhipment at sea allowing catches of multiple fishing vessels to be combined before landing, making the tracing of fish back to individual vessels currently impossible[47]. A lack of consistent, accurate and transparent data from the point of capture to its final destination means that seafood caught illegally or unethically can effectively be laundered by combining it with legally caught fish in subsequent processing steps. The large consumer markets of the global north, including the USA and Europe, import large volumes of seafood to supplement domestic supply. Given that, for example, up to 32% of wild-caught fish imported into the US is estimated to have been caught illegally[48], it seems likely that fish caught under conditions of modern slavery can also enter the domestic supply chains of countries otherwise considered low risk for labour issues in fishing.

Kittinger et al.[49] called for the research community to more explicitly recognise and address the social dimensions of the ecological crises in the oceans. Modern slavery at sea is such an issue, but there is currently a paucity of quantitative research. The global data on country-level slavery from the GSI[45] and comprehensive data on fisheries and seafood trade from the *Sea Around Us*[1,50] and the United Nations' COMTRADE database provide a base for a preliminary investigation. Here we (1) examine the empirical relationship between the GSI's country-wide prevalence of modern slavery (in all aspects of a country's economy) and fisheries' governance and financial performance; (2) separately identify factors common to those countries with reported labour issues specific to fisheries; and (3) model potential consumer exposure to modern slavery-derived seafood products by quantifying the flows of fish from high (GSI-based) slavery risk environments to relatively lower slavery risk markets.

## Results

**Analyses**. Our analyses were performed in three separate stages. The first used linear models to test the overall relationship between the national prevalence of modern slavery, across all aspects of a country's economy, and industrial fisheries attributes among the major fishing countries of the world. Country-level estimates of the overall prevalence of modern slavery (of all types and across all economic aspects of a country) were taken from the GSI[45], and fisheries catch and economic data were obtained from the *Sea Around Us*[1,50]. Here national-level GSI data covering all socio-economic aspects of a country were used as a proxy for likely fisheries-specific estimates of slavery prevalence, which are currently lacking for fisheries at the global level. The second stage used a multivariate clustering approach to identify additional fisheries and economic factors shared by countries with specifically identified slavery issues in fisheries, as reported in the literature and media; this second analysis did not use GSI data. The goal was to develop a qualitative risk model based on the fisheries and socio-economic factors associated with reported incidents of slavery that can frame further research efforts. The third analysis used United Nations' COMTRADE data and the GSI slavery prevalence measure to model the impact of the global trade in seafood on the presence of potentially slave-caught or processed seafood in consumer markets in the United States and Europe, regions where the risk of slave-produced seafood in domestic fisheries is otherwise considered low.

**Country-level slavery and fisheries metrics**. Linear regression modelling focused on the 20 highest-volume fishing countries, collectively landing over 80% of global industrial fisheries catch. Exploratory analysis found the best explanatory variables to be percentage of unreported catch and landed value of catch (Supplementary Table 1). The mapping of unreported catch (Fig. 1a), mean landed value of the catch (Fig. 1b), and the overall prevalence of modern slavery at the country level (Fig. 1c) for the world's major fishing countries suggest regional hot-spots of forced labour or modern slavery in Asia, Sub-Saharan Africa and parts of South America. Generally, these are areas with relatively high levels of unreported catch, predominantly low value fisheries and a relatively high overall prevalence of modern slavery at a national level. The country-wide prevalence of modern slavery in a given country is positively correlated with higher levels of unreported catch ($R^2 = 0.24$, $p = 0.017$, Fig. 1d) and negatively correlated with the landed value per tonne of fish being caught ($R^2 = 0.26$, $p = 0.013$, Fig. 1e). The multiple linear regression model using both variables explained 46% of the variance in the overall prevalence of country-wide modern slavery among countries ($p < 0.01$, Fig. 1f). Thus a high level of unreported catch,

representing poor management or enforcement oversight of fisheries, and a low unit-value catch, indicating poorer profitability, all other things being equal, correlate with a higher prevalence of modern slavery in the general economy of that country (Fig. 1f). While correlation is not causation, these results suggest a link between the presence of slavery and the overall performance of a country's fisheries. The analysis suggests broad underlying trends, yet also identifies outliers whose fisheries performance and country-level modern slavery prevalence do not fit the overall trend. While caution is needed when making inferences about specific economic sector-level labour abuses from the country-level GSI, the present analysis provides a basis for further, detailed sector-specific investigation.

**Risk factors associated with known labour abuses at sea**. Having identified in the first analysis a broad correlation between the prevalence of modern slavery at the country level and two key fisheries attributes (unreported catch and mean landed value) for the top 20 fishing countries, we performed a separate principal component analysis (PCA) for the same 20 countries. The PCA grouped countries across six variables describing their economic status and fisheries performance/policy: unreported catch (% Unreported), percentage of catch caught outside their own exclusive economic zone (EEZ) (% Catch outside EEZ), per person Gross Domestic Product (GDP per capita; www.imf.org), level of harmful subsidies as a percentage of landed value (% Subsidy), mean landed value per fisher (Value per fisher), and mean distance of catch (Distance). No GSI data were used for this analysis. PCA summarises information contained in a group of *n* predictor variables as *n* principal components that capture the main dimensions of variation among the groups being measured, in this case the top 20 fishing countries. The first two components of the PCA explained 74% of the variation between countries. The first principal component axis (PC1) explained 44% of variance between countries and was correlated most strongly with '% Subsidy', '% Catch outside EEZ' and 'Distance'. The second principal component axis (PC2) explained a further 30% of variance and was correlated positively with '% Unreported' and negatively with 'GDP per capita' and 'Value per fisher (Fig. 2). Overall, the individual explanatory variables made similar contributions to the model (Supplementary Figure 1). Clustering countries based on their score (i.e. location) on the first two PCA dimensions divided them into three distinct groups (Fig. 2). The first cluster comprised seven countries (red in Fig. 2), most of which have been reported for or suspected in serious labour abuses on fishing vessels[15,32,39,40,51,52]. Countries with documented incidents of serious labour abuses in fisheries are therefore characterised by high levels of unreported catch ('% Unreported'), a high proportion of catch taken outside their own EEZs ('% Catch outside EEZ') at a greater distance from home waters ('Distance') and higher than average levels of harmful subsidies ('% Subsidy'). It appears that distance from home waters, non-EEZ fishing and poor fisheries oversight ('% Unreported') may substitute as potential risk factors for modern slavery in fisheries. However, owing to a lack of fisheries specific data on modern slavery by country, such conclusions must be drawn with caution and require further investigation.

The second group of countries (orange in Fig. 2) included mainly South American and Asian fishing countries with largely domestic fisheries or fisheries that use the waters of immediate neighbours. These countries were characterised not only by low levels of fishing outside their own or immediate neighbours' EEZs ('% Catch outside EEZ') and low levels of harmful subsidies ('% Subsidy') but also relatively low GDP per capita ('GDP per capita') and low value fisheries ('Value per fisher'). Future

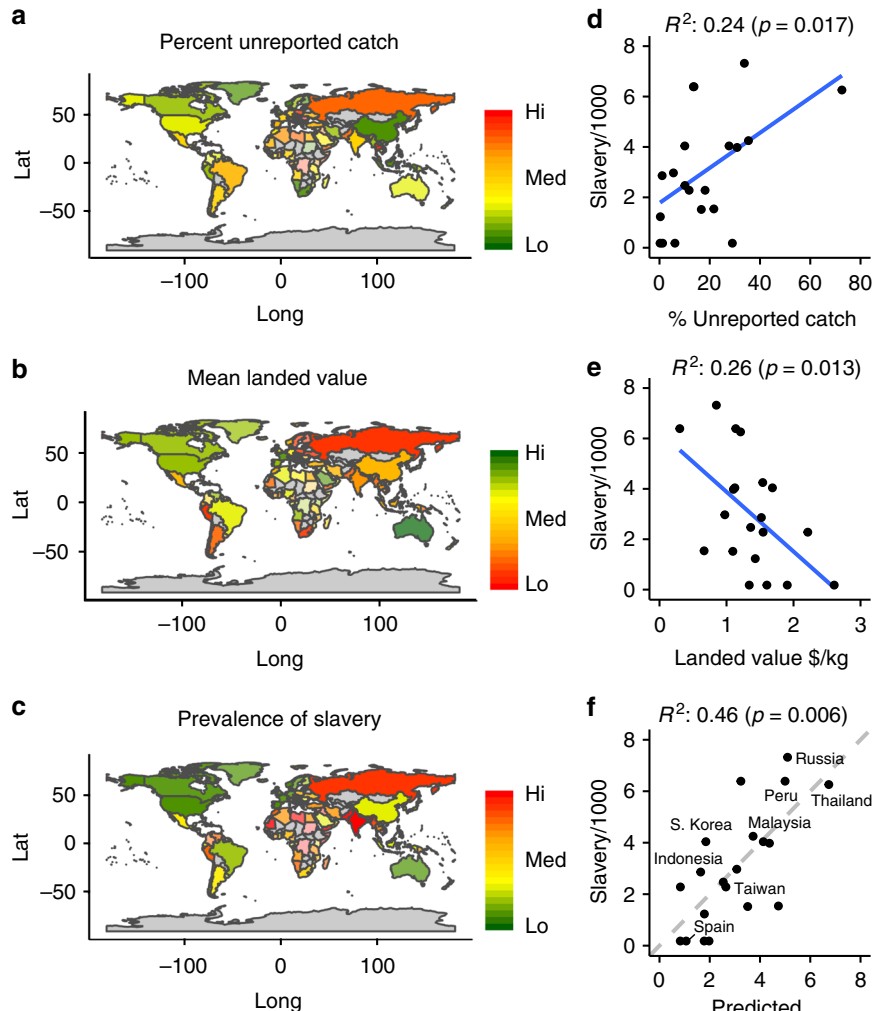

**Fig. 1** Global patterns in country-level slavery and fisheries catch and value. Maps show **a** percentage of unreported catch, **b** mean landed catch value per kg, and **c** national prevalence of slavery, colour-coded by country. Scatterplots show relationship between country-level slavery prevalence and individual fisheries variables for the 20 largest fishing countries: **d** prevalence of slavery per thousand people (Slavery/1000) vs unreported catch (% Unreported catch), **e** prevalence of slavery vs mean landed value (Landed value $/kg), and **f** observed against predicted values for a combined model, with selected European, Asian and South American countries labelled. Regression model $R^2$ values and $F$-test $p$ values are labelled on scatterplots

research may show how these countries and these fisheries parameters relate to potential labour abuses or modern slavery in fisheries. The third group (green in Fig. 2) consisted of countries generally deemed low slavery risk (the USA and three European fishing countries) that were associated with low levels of unreported catch ('% Unreported'), high GDP per person ('GDP per capita') and high landed value per fisher ('Value per fisher').

**Global trade and slave-produced seafood.** Finally, we assessed seafood trade data in relation to modern slavery risk to understand the extent to which fish being caught and processed by high slavery-risk countries is potentially consumed in markets that have a low risk of slavery in their own domestic supply chain. Globally, an average of >33 million tonnes of seafood were traded annually between 2005 and 2014, based on harmonised UN COMTRADE data (http://www.cepii.fr). Seafood supply in the top developed countries includes significant proportions of imported wild-caught fish: in the United States, around 45% of domestically consumed seafood is imported wild-caught fish

(http://www.nmfs.noaa.gov), while in the EU this is 50%[53]. Total imports are even higher when aquaculture products are considered. Consequently, the seafood available to consumers in these otherwise low slavery-risk countries can end up being a mix of domestic products from local fisheries, predominantly in national waters, and products imported from a wide variety of other countries, including from countries with a higher risk of country-wide slavery.

The United States is highly dependent on imported seafood to meet domestic demand and accounts for roughly 14% of global seafood imports. It has a national slavery prevalence of 1.8 victims per 10,000 persons in the population (0.018%)[45]. Expressed in term of kilograms of potential slavery-risk seafood per tonne, this equates to a slavery risk of 0.2 kg per tonne of domestically produced seafood, assuming the national prevalence of slavery is applied to all sectors of the seafood industry. Based on the average volumes of seafood imported from other countries, in particular from Asia-Pacific countries, seafood imported into the US has an average potential slavery risk of 3.1 kg per tonne, 17 times higher than the risk of seafood sourced from domestic fisheries (Fig. 3a). After accounting for the mix of domestic and imported seafood in

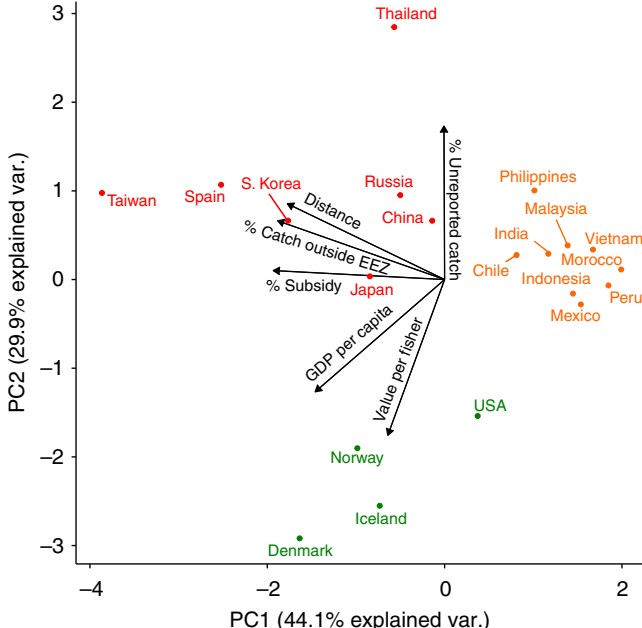

**Fig. 2** Biplot of principal components analysis (PCA) for the top 20 industrial fishing countries. Countries are represented based on their aggregate scores across three economic and three fishing activity measures. Arrows indicate direction of increasing value for each variable. Colour-coding indicates cluster membership determined by k-means clustering of countries based on their scores on the main PCA dimensions (PC1 and PC2)

US domestic supply, the potential slavery risk of seafood supply within the United States increases 8.5 times due to its dependence on imports (Fig. 3a).

Similarly, the low slavery-risk countries of Europe also account for 14% of global seafood imports. Based on the GSI assessment, these countries (i.e., Denmark, France, Germany, Ireland, Netherlands, Norway, Spain, Sweden and United Kingdom) have an average national slavery prevalence of 2.8 victims per 10,000 persons (0.028%) across their combined populations. Considering the slavery prevalence of the countries from where seafood is imported into this block, the potential slavery risk of imported seafood is 3.8 kg per tonne, 13 times higher than that for their domestically sourced seafood (0.3 kg per tonne). Thus the mix of imported and domestically sourced seafood increases consumer exposure to potentially slavery-derived products is 8.6 times (Fig. 3b), similar to the modelled effect in the United States.

## Discussion

Sustainable fisheries underpin both environmental and socio-economic development goals for the oceans[54], but until recently much of the research has focused on environmental and economic impacts, with less focus on human rights[4,29]. While links between modern slavery and environmental destruction in illegal mining and deforestation are now well recognised[55], the connections between environmental challenges and human rights in fisheries have been less systematically documented. However, labour issues in fisheries have received increased attention in recent years[14,15,25,56], leading to emerging responses from governments and trading partners (e.g. Thailand-EU), non-governmental organisations (NGOs, e.g. Fair Trade), and major industry–research partnerships such as the Seafood Business for Ocean Stewardship initiative (SeaBOS)[57,58]. An understanding of potential slavery at sea at the global level can place these isolated cases and responses in a broader policy context.

The present analyses have focussed on using comprehensive and publicly available global data sets to examine empirical links between country-level slavery prevalence and industrial fisheries and the role of the global trade in seafood in moving seafood products from potentially high slavery-risk producer to low-risk consumer countries. Treating the national, non-fisheries-specific prevalence of modern slavery measured by the GSI[45] as a proxy for the as-yet unmeasured slavery risk across fishing industry sectors, we found a correlation between the prevalence of modern slavery within a country and proxies for poor fisheries accountability (i.e., high levels of unreported catch) and low profitability (i.e., low landed value of the catch) in the industrial fisheries of the major fishing countries. It should be emphasised that the GSI is not currently designed to differentiate sector-specific slavery risks, such as for fisheries. Indeed, localised fisheries-specific surveys conducted by NGOs suggest that the national, country-level GSI measure used here may in fact underestimate modern slavery practices in some industrialised fishing fleets. For example, interviews with migrant fishers in Thailand found that 17% of respondents had experienced conditions of modern slavery[59], compared with the GSI's estimate of <1% of workers nationally across all sectors. Conversely, for countries where land-based slavery practices dominate (for example, mining or agriculture), the GSI's estimate may imply a higher risk for fisheries than may be the case. With this caveat, there remains a broadly linear relationship between national, country-wide levels of slavery prevalence and poor fisheries performance, based on the global data currently available.

To explore risk factors linking the smaller subset of known incidents of slavery at sea, a separate multivariate analysis was then used to identify fisheries and economic attributes shared by those countries with documented fisheries-specific labour abuses. Cluster analysis indicated that countries with documented labour abuses in sections of their fishing industry share several key features: high levels of harmful capacity-enhancing subsidies, likely leading to excess fishing capacity, increased competition and reduced per-vessel profitability; low catch value per individual fisher, suggesting downward pressure on wages; high levels of undocumented fishing activity, implying poor monitoring and enforcement of vessel operations at sea; and a reliance on fishing far from home in the waters of other countries where regulatory violations may be more likely to go undetected by domestic agencies. Additional evidence of the role of distant-water fisheries in slavery at sea appears in reports detailing specific cases of labour abuse in fisheries, with many victims never even visiting their employer's country (i.e. the vessel's flag or beneficial ownership state), instead transiting through maritime hubs or countries closer to fishing grounds[15,51]. The nature of distant-water fishing operations, where transhipment of catch and crew at sea are commonplace and observer coverage is typically low, appears to facilitate illegal behaviour[47]. The last factor in our multivariate model, GDP per capita, may reflect the importance of economic disparity between labour demand and labour supply countries in driving labour migration, with documented incidents of slavery occurring in countries with relatively high per capita wealth compared to the country of origin of the victims[59]. For example, Thailand's GDP per capita is over three and four times that of Myanmar and Cambodia, respectively, i.e. countries from which it sources the majority of its foreign fishing labour (www.imf.org)[60]. In drawing these conclusions from our analyses, we recognise that fisheries within a single country will differ widely on both social and environmental performance metrics, as the coexistence of Fair Trade-certified tuna fisheries (www.FairTradeUSA.org) and fishing slaves trapped on islands in Indonesia[26] demonstrates. Nevertheless, while such distinctions must be factored into domestic policy, a model of the common drivers of potential

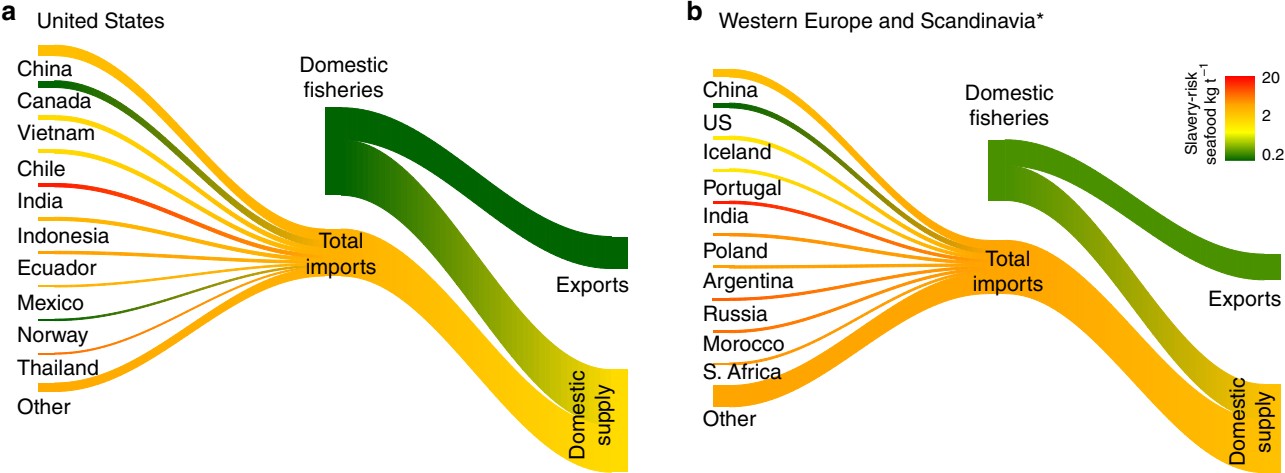

**Fig. 3** River plots showing the impact of seafood imports on the modern slavery risk of domestically consumed seafood. Slavery risk is expressed in kilograms of seafood from slavery-risk countries per tonne consumed. Slavery risk scores based on the Global Slavery Index; trade flows from CEPII's BACI database of harmonised UN COMTRADE data. Plots show seafood imports for **a** the United States and **b** Western Europe and Scandinavia (includes Denmark, France, Germany, Ireland, Netherlands, Norway, Spain, Sweden, and United Kingdom). Colour of trade flow components indicates the intensity of slavery risk

slavery at sea across fishing countries can provide a framework to prioritise research and policy development at the international level. While exploratory in nature, our findings suggest that the well-recognised subsidy-fuelled race to fish, a lack of adequate monitoring, control and surveillance of industrial fishing activities and the influence of economic disparity on labour markets has made this sector a fertile ground for modern slavery and other violations[24,61,62].

The volume, diversity and global scale of the international trade in seafood[46] means that seafood produced by countries with poor records in both modern slavery and fisheries governance may find its way into the domestic markets of better regulated countries. Potentially slave-caught or processed seafood can reach consumers directly, as wild-caught product, and indirectly via fishmeal used in livestock and aquaculture feed. Fishmeal supplied by reduction fisheries targeting pelagic fishes, together with millions of tonnes of unmarketable trash fish caught as bycatch, eventually end up on consumer plates as farmed salmon, tuna or prawns or even pork, chicken, eggs or beef[63,64]. Many wealthy seafood producing countries, including the United States and European countries, export much of the fish produced by their own fisheries and meet net domestic demand with imports of cheaper seafood products from areas such as Southeast Asia, Africa and Russia[65,66]. Our analysis of UN trade data suggested that this could result in a greater than eight-fold increase in the exposure of their consumers to potentially slave-caught or produced seafood. To date, however, cases linking specific products to labour abuses have been isolated, and further work on traceability as well as fisheries slavery is required to confirm this hypothesis. For comparison, work done to model the flow of illegally caught seafood into the major consumer markets of the US and Japan (together almost 30% of global seafood imports) found that illegally caught products likely constituted 20–32% and 24–36%, respectively, of each country's wild seafood imports[48,67]. It seems plausible that the current lack of supply chain transparency and product traceability that allows the products of illegal and unreported fishing to enter supply chains also facilitates the international movement of slave-caught and processed seafood.

The issues raised by our modelling of slavery, fisheries and seafood trade suggest four broad areas of policy engagement: (1) regulation and enforcement, specifically universal minimum standards for crew pay and conditions, such as those specified in the International Labour Organisation's Work in Fishing Convention (C-188), and improved monitoring and enforcement of currently weak jurisdictions, including the high seas, to reduce the scope for unsustainable and unethical fishing practices[68]; (2) supply chain transparency, specifically by adopting supply chain legislation, such as the UK's Modern Slavery Act (Modern Slavery Act 2015, s 54), which can bolster industry-led efforts such as SeaBOS to leverage businesses' market position to tackle sustainability and ethical issues[58]. Policing supply chains can be supported by technologies, such as Blockchain ledgers and smart seafood labelling, which improve the security and lower the cost of reliable supply chain data[69]; (3) industry restructuring, specifically by reducing harmful subsidies that currently overcapitalise fishing capacity[5,70], and redirecting subsidies towards enforcement and the rebuilding of sustainably managed small-scale fisheries capable of providing more and better livelihoods[11,71]; and (4) improving equity between stakeholders in fisheries, specifically by restricting high seas fishing, which is currently dominated by higher-income countries[72]. Complete closure of the high seas to fishing has been modelled to reduce income inequality among fishing countries by 50%, by ensuring more equitable access to valuable migratory fish stocks[73].

These issues have also emerged as key topics in the broader discussions of sustainability in global fisheries as they affect our current ability to effectively manage fisheries for the collective benefit of humanity. This apparent overlap offers an opportunity to leverage regional and international initiatives to benefit both ecological sustainability and social/ethical goals. As research around labour issues in fisheries crystallises, there is great potential for marine scientists and social scientists to collaborate in developing policy frameworks that jointly tackle sustainability and human rights issues. The rapid expansion of industrialised fishing over the past 60+ years has negatively impacted the ability of marine ecosystems to sustainably supply humanity with seafood. The concurrent failure by government decision makers, policy developers and fisheries managers in many regions to adapt to the changes in industrial fisheries has rendered much of the high seas, as well as the waters of developing countries in fisheries-rich areas such as West Africa, open to abuse of both fisheries regulations and international labour standards, allowing illegal fishing and, potentially, labour abuses to flourish[15,24,73,74].

Modern slavery and fisheries' performance appear linked at the international level, with a correlation between increased prevalence of country-level modern slavery and higher levels of unreported catches and lower mean value of the catch of industrial fisheries for the 20 countries who supply the bulk of the world's wild-caught seafood. Further research and improved data are urgently needed, as the GSI can presently only report on the risk of slavery at the whole-country level. Given the current lack of reliable data on the prevalence of fishery-specific slavery and labour abuses, the country-level GSI is the most appropriate substitute metric currently available. Based on the limited information available on specific instances of slavery at sea, the over-subsidised and often poorly governed, distant-water fishing fleets of higher-income countries may be at particular risk of labour abuses and modern slavery. Our preliminary trade model, using peer-to-peer trade in seafood products, indicates that products of fisheries from slavery-prone regions/countries may be consumed in developed countries in significant quantities, potentially making seafood consumers in developed countries unwitting participants in modern slavery.

Much additional work is required to quantify the prevalence of labour abuses and modern slavery in seafood capture, aquaculture, processing and in the seafood supply chain. Generating comprehensive and accurate estimates of the prevalence of modern slavery in the fishing industry and seafood supply chain will not be easy, as fishing vessels rank among the world's most inaccessible workplaces. However, like the challenge of enforcing environmentally more benign fishing practices, it is an obstacle that must be overcome.

## Methods

**Data sources**. Data on global fish catches by fishing country were obtained from the *Sea Around Us* reconstructed global catch database[1]. The methods used for catch data reconstructions and the spatial allocation of global catches are well established[75] and individual country reconstructions are summarised in Pauly and Zeller[50], with detailed technical descriptions accessible via www.seaaroundus.org for each country. Using the *Sea Around Us* reconstructed catch data, we calculated the annual mean (±SE) reported and unreported industrial landings (in tonnes, excluding discarded catch) for the decade between 2005 and 2014 for the top 20 industrial fishing countries representing 80% of global landings. Thus here the term catch is used to represent landed catch (i.e. landings) and excludes discarded catch[76]. In line with international data reporting mechanisms, all catches are supposed to be reported by the flag-state of the fishing vessel (i.e., the flag flown by the fishing vessel) and not the country of residence of the beneficial owner. The fishing activity modelled in our analysis is therefore that of the flag-state reporting the catch on behalf of its flagged fleets. Clearly, flag-hopping, i.e. the tendency by some distant-water fleets to regularly and often rapidly re-register to different flags, makes data reporting for distant-water fleets challenging, and better resolution of this issue needs to be a subject of further investigation.

Data on fisheries employment in the industrial sector used here were taken from Teh and Sumaila[11], excluding small-scale fisheries. Estimates of fisheries subsidies by category (beneficial, harmful and ambiguous) and type (fuel, vessel buyback, etc.) were obtained from the *Sea Around Us*[5]. Estimates of GDP per capita, in purchasing power parity-adjusted US dollars, were obtained from the International Monetary Foundation's IMF DataMapper site (https://www.imf.org/external/datamapper/PPPPC@WEO/OEMDC/ADVEC/WEOWORLD). The catch weighted mean distance of fishing activity from home for each country was calculated using the ½×½ degree cell-allocated catch data of the *Sea Around Us*[75]. *Sea Around Us* catch data are spatially allocated by intersecting biological probability distributions for each taxon in the catch data with a global fishing access database detailing in which country's EEZ foreign fleets are permitted or have been observed to fish[75]. Distance from home for each catch cell was calculated as the great circle distance between the centroid of each catch cell and the closest domestic port of the fishing country, with port locations taken from the World Ports Index. The catch weighted mean distance was the weighted average of all such cell-port distances, weighted by the catch for that country in each spatial cell, using the methodology employed in Tickler et al.[77].

Data on the scale of modern slavery were taken from the GSI database[45], which reports estimates of vulnerability to and prevalence of slavery for 167 countries. Modern slavery was defined as 'situations of exploitation that a person cannot refuse or leave because of threats, violence, coercion, abuse of power or deception'[45]. Slavery vulnerability scores in the GSI were generated based on a detailed model of country-level measures of governance and civil protections[45]. Prevalence, defined as the percentage of the population trapped in modern slavery,

was estimated from data collected on behalf of the WFF as part of the Gallup World Poll (www.gallup.com) through face-to-face interviews with >42,000 respondents in 25 countries between 2014 and 2016. Estimates for unsurveyed countries were extrapolated from the subset of surveyed countries using a model based on the relationship between prevalence and vulnerability[45]. Slavery prevalence was presented in this study as individuals per 1000 population rather than a percentage for ease of comprehension and represents country-wide slavery prevalence across all economic sectors and not fishing-sector-specific slavery. A detailed description of the methods used for measuring modern slavery is provided in the 2016 GSI[45] and the references therein.

Global trade flows for seafood commodities, estimated as imports and exports of individual seafood commodities in tonnes of seafood product (not wet weight) by country, were taken from the BACI harmonised trade database provided by the Centre d'Etudes Prospectives et d'Informations Internationales (CEPII) in France (www.cepii.fr). The BACI database uses data from the UN's COMTRADE database, processed so as to resolve inconsistencies between commodity-level import and export volumes and values between countries. BACI data categorised by commodity using the 2012 harmonised system six-digit codes were used, wherein the group of commodities beginning with 03---- represents both wild-caught and farmed seafood products; it was not possible to distinguish between farmed and wild-caught products. The BACI estimates of trade flows were averaged for 2011–2014.

**GSI and fisheries performance measures**. The relationship between country-wide slavery prevalence and candidate fisheries measures (percentage of unreported landings, landed value of catch per kg and tonnes landed per fisher) was tested using multiple linear regression, with competing models compared using sample size corrected Akaike's Information Criteria (AIC) scores (AICc). Model data were taken from the top 20 industrial fishing countries, representing 80% of global catch. Given the high prevalence of land-based modern slavery in India[45], our approach was to treat India as an outlier for the linear regression analysis. This decision was made based on additional information available for India, for which GSI data were collected at the state level, indicating that modern slavery levels in land-locked states heavily influenced the whole-country estimate. The best model, judged by AICc, used percentage of unreported landings and landed value of catch per kg as predictor variables (Supplementary Table 1). The relationships between country-wide slavery prevalence and percentage of unreported catch and between country-wide slavery prevalence and the mean landed value of catch were visualised in individual scatterplots. Model fit for the final model was visualised by plotting observed against predicted values.

Sensitivity analysis was performed on the final multiple regression model to test the effect of uncertainty in the fisheries and slavery estimates on the model outcome. Fisheries parameters were modelled for each country as being normally distributed with the mean and standard deviation calculated from the 2005–2014 *Sea Around Us* data. Country-wide slavery data were modelled as normally distributed with a mean equal to the reported value and standard deviation equal to the 95% confidence interval divided by 1.96. A Monte Carlo simulation of 10,000 model runs of the multiple linear regression model was used to build a distribution of $R^2$ values based on likely values for model inputs. Histograms of the output for three alternatives were plotted: varying all variables, varying only fisheries variables and varying only slavery variables (Supplementary Figure 2). The median $R^2$ value for models varying all variables was 0.29, vs 0.46 for the model using mean fisheries values and the GSI-reported country-wide slavery values, which is reported in the results.

To visualise global geographic patterns in both country-wide slavery and fisheries performance, fishing countries' mean values for the predictor and response variables used in the final model (percentage of unreported catch, landed value of catch per kg, and slavery prevalence at the national level) were mapped. Countries were classified by the three measures, with red representing poor performance (high unreported catch, low mean landed value, high country-wide slavery prevalence) and green the opposite. The classification of prevalence of modern slavery, as reported in the GSI, are country-wide data and not specific to the fisheries sector.

**Modelling risk factors associated with slavery at sea**. PCA followed by $k$-means clustering was performed on the top 20 fishing countries based on 6 measures hypothesised to predict the occurrence of modern slavery in fisheries: unreported catch ('% Unreported'), mean landed value per fisher ('Value per fisher'), percentage of catch caught outside their own EEZ ('Catch outside EEZ'), GDP per capita (www.imf.org), level of harmful subsidies as a percentage of landed value ('% Subsidy'[5]) and mean distance of catch ('Distance') calculated from cell-level catch data of the *Sea Around Us*[1,75]. The objective of the analysis was to identify the shared characteristics of groups of major fishing countries based on their involvement in known cases of modern slavery in fisheries, to explain outliers in the linear model and to identify other at-risk fisheries that were not highlighted by the linear analysis. Scores on the first two principle components of the PCA, capturing the most important components of variation in the predictor data set, were used to group the countries using a $k$-means clustering algorithm (i.e. grouping countries into $k$ groups based on their similarity across the composite measures). The optimum number of clusters ($k$) for this step was determined analytically using the

NbClust() function in R, which finds the number of clusters that minimises the total within-cluster variance (i.e. makes the group members as alike as possible). The first two components of the PCA were visualised as a biplot, with the cluster members colour-coded (red, orange, green) based on their score on the first two PCA components.

**Slavery and global seafood trade**. The impact of imports of seafood into a country or region on the country-wide slavery prevalence (risk) associated with its domestic seafood supply was modelled using commodity-level country-to-country trade flows in the BACI harmonised UN COMTRADE data. The BACI data allow individual commodity flows between countries to be identified, so that flows of seafood carrying different slavery risks, based on country of production, can be precisely estimated. No distinction could be made between seafood caught by a country and exported, or imported, processed and re-exported, since that level of information is not supplied. However, this was not a significant issue since national cross-sectoral country-wide slavery prevalence was being used to score seafood exported from a country. Therefore, it was implicitly assumed that all seafood exported by a given country, whether caught by domestic fleets or processed from imports, carried the same risk of potentially involving slavery. The slavery prevalence of seafood imports into a particular country or group of countries was then calculated as average of the GSI country-wide slavery prevalence scores of the countries supplying that seafood, weighted by tonnes of seafood products imported from each country. Although the GSI slavery prevalence is not specific to the capture fisheries sector, traded fisheries products necessarily involve labour across multiple sectors beyond fisheries, and so a cross-sectorial estimate of the prevalence of slavery gives a reasonable estimate of the slavery risk of products originating in or being re-exported from a particular country. Domestic supply in turn was the average of the slavery prevalence of imports and domestic production, weighted by import tonnage and domestic production net of exports. Internal trade within a bloc of importing countries was considered part of domestic supply, rather than exports. Seafood trade and consumption flows were visualised using a Sankey plot (also known as a river plot) where the width of connections between nodes is proportional to tonnes traded or produced. River plots were produced in this way for the United States (14% of global imports) and the low slavery risk seafood-importing countries of Western Europe (Denmark, France, Germany, Ireland, Netherlands, Norway, Spain, Sweden and United Kingdom; 14% of global imports).

All statistical analyses were performed using the R statistical language and packages in R Studio.

## Data availability

All relevant data are available on request from the authors. All *Sea Around Us* data are freely available via www.seaaroundus.org and can also be accessed via the R package *seaaroundus* (see https://github.com/seaaroundus/). Teh & Sumaila's fisheries employment estimates are available at https://onlinelibrary.wiley.com/action/downloadSupplement? https://doi.org/10.1111/j.1467-2979.2011.00450.x&file=faf450_sm_Table.S1.doc. Country-level estimates of the prevalence of modern slavery were taken from the Global Slavery Index (https://www.walkfreefoundation.org/). Global trade flows for seafood commodities are provided by the Centre d'Etudes Prospectives et d'Informations Internationales (CEPII) (http://www.cepii.fr). The economic data used can be obtained from the International Monetary Foundation's DataMapper site (https://www.imf.org/external/datamapper).

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

## Acknowledgements

D.T., J.J.M., D.P., U.R.S. and D.Z. acknowledge the support of the *Sea Around Us*, the Fisheries Economics Research Unit and the *Sea Around Us – Indian Ocean*. All *Sea Around Us* activities are supported by the Marisla Foundation, the Paul M. Angell Family Foundation, the Oak Foundation, the MAVA Foundation, the David and Lucile Packard Foundation and Oceana. K.B. F.D., E.G., J.J.L., B.O. and D.T. acknowledge the support of the Walk Free Foundation.

## Author contributions

J.J.M., J.A.H.F., F.D., D.P., D.T. and D.Z. were involved in the conception of the project and bringing together the Walk Free Foundation and *Sea Around Us* databases. D.T. performed all data analyses and figure preparation, supported by E.G., J.J.L. and B.O. who collated and prepared the data sources and performed supporting analyses. D.T., J.J.M., F.D. and D.Z. drafted the manuscript, with substantial editorial input from K.B., J.A.H.F., E.G., B.O., D.P. and U.R.S.

## Additional information

**Competing interests:** The authors declare no competing interests.

