## [Peer Review File · Nature Communications]

Reviewers' comments:

Reviewer #1 (Remarks to the Author):

This is a novel, important contribution. I appreciate your analysis, drawing on a series of big 'n' data sets that cover global data on fish catch, trade and modern slavery, to really highlight the challenges of working conditions in off-shore fishing vessels. Your visuals are compelling and easy to follow for a non-stats expert (me!). I applaud your timely work.

My comments are linked to your framing and conclusions, where I'd like to see further nuance and analysis, that can be drawn out from your findings.

Framing: The concept of modern day slavery is appealing (and for men facing the worst working conditions this term is appropriate) and framing your paper in this way also makes for an explicit link to the Modern Slavery Index. However, you can go further than this, acknowledging that a range of poor working conditions exist within the fisheries (from human trafficking to practices that may be poor but that men agree to and negotiate). The idea of a continuum of unfree labour is helpful here. This is to also show that there are cases where men exert agency and to not only pain them as victims. It is also worth pointing out how little unpacking of labour abuse in fisheries exists, and this is where migration scholars have done some thoughtful analysis.

Discussion and conclusions. I would like to see you go further here. For example, if laws are to be enforced perhaps part of this is to insight that ILO C-188 is ratified. I also think you can add more detail on the Thai case as NGOs, IOs and the business community also play a strong role in the significant progress (to show it is not only New Zealand but also global South countries who are capable of thinking this through – Vandergeest et al. 2017 provide a brief overview of how Thai NGOs have been involved, and I agree that we don't know if this will be sustainable given it is a Military government).

Also, in terms of supply chain accountability, perhaps add some more detail in. For example, certification generally has focused on global North fisheries and even here there are ecological issues. Even if MSC was to add in labour issues (which they are not in doing in any significant way), they are not really certifying global South fisheries anyways. How do companies insist on supply chain accountability and traceability given that serious labour abuse occurs in the global South (but not only)? While companies such as Thai Union have all seafood operations go in-house, in a sense creating sustainability enclaves for global North consumption, how do we ensure basic human rights for fish labourers in addition to insisting on more sustainable fishing practices? Is FairTrade Tuna a model to follow? <http://3blmedia.com/News/Safeway-and-Fair-Trade-USA-Launch-Worlds-First-Fair-Trade-Certifiedtm-Seafood> I realize there are not any easy answers here, but more to see you probe a bit more as your writing team has considerable depth on these issues.

Minor detail: Refs 29-33 really refer to Burmese men working in Thailand, Philippine men trafficked across a number of industries, and Burmese and Khmer men on Thai boats in Indonesian waters etc. Little has been flagged in terms of Burmese fisheries, for example. You can easily update / add nuance to this section, drawing on more recent reports, to signal labour abuse scandals emerging in Thailand, Taiwan (the Greenpeace 'Turn the Tide' report looks at labour abuse in Taiwanese fisheries), Ireland or Hawaii just in 2014-2016 while Christina Stringer's work in New Zealand is helpful for understanding the role of Indonesian workers on Korean vessels in the early 2010s.

Other papers to add to your analysis:

Lewis & Boyle, 2017. The Expanding Role of Traceability in Seafood. *Journal of Food Science*.

Kittinger et al. 2017 Committing to Socially Responsible Seafood. *Science*.

Reviewer #2 (Remarks to the Author):

REVIEW

Manuscript #: NCOMMS-17-25933

Title: Modern slavery facilitates overfishing in global fisheries

General Comments:

The authors are to be commended for undertaking an ambitious analysis assessing the relationship between overfishing and modern slavery. The study provides an important contribution to our understanding of this important issue and I believe it is worthy of acceptance, following some minor revisions.

I have several general comments and a list of specific comments, which if addressed I believe will strengthen the paper.

First, the authors make the point that there our global understanding of the scale of this issue in the seafood sector is quite limited. The Global Slavery Index is one of the only global level indicators available, but it is not focused specifically on the seafood industry and it's survey-based methodology has some limitations (as noted in line 228+). I think the relationship the authors have found certainly highlights the co-occurrence of these issues, but without stronger evidence I don't believe at this point you can say definitively that slavery facilitates overfishing. The authors present a good set of arguments and lines of reasoning for how this relationship can work, but without definitive analysis and evidence of the subsidy effect I think at this point you are pointing out two main (and still very important) issues:

- Overfishing and modern slavery co-occur in areas with poor governance which likely mediates both and there is strong potential for poor labor practices to provide subsidy effect for fisheries
- Products from these geographies are flowing into markets in the EU and US which has serious implications for trade as well as for the solutions required to address these issues

If the authors are still very keen on making the link on the subsidy effect, one suggestion is to assess how many fisheries would become uneconomic if labor practices were just and human rights were fully protected. How much of the overfishing problem would this fix?

Second, it is important to note at the outset that slavery and associated human rights and labor violations are among the most egregious practices globally in the seafood sector. However, they are not the only issues. Social sustainability can mean a lot of different things to different groups. A recent effort (Kittinger et al. 2017) focused on generating a global consensus definition of social responsibility for the seafood sector. This initiative took a similar approach as was taken with environmental sustainability parameters, which derived primarily from the FAO's Code of Conduct. Similarly, the social responsibility framework derived primarily from the FAO's Voluntary Guidelines for Small-scale fisheries, which like the code of conduct was developed globally with significant stakeholder input. I recommend the authors at a minimum provide some context on these other issues, while keeping their analytical focus on the analysis of human rights and slavery.

Third, the authors risk-based approach is a good one but it can be problematic at the country level. For a country like Indonesia, for example – where known labor violations and slavery have occurred – this means that a country-level rating obscures a wide range of performance from good to egregious. This has the effect of painting all the fisheries in a complex geography with a single risk rating, which is problematic both for well-performing fisheries (socially and environmentally) who may not be rewarded in the marketplace, as well as the most poorly performing fisheries, which may be actually, require immediate attention, and may be much worse than a single risk rating could convey. Also – the risk-based analysis at the country level can incorrectly convey that

this is a problem that needs to be addressed via domestic policy in each of these geographies. However, due to the high degree of migrant labor in fleets, actions almost certainly require better policy and enforcement within the country in which the violation occurs (or in the case of DWFN, on vessels flagged in these countries) but also the source country for the laborers.

Fourth, I recommend the authors acquaint themselves with the UN Guiding Principles of Business and Human Rights (also known as the “Ruggie principles”; http://www.ohchr.org/Documents/Publications/GuidingPrinciplesBusinessHR_EN.pdf). This well-known guidance document describes the clear corporate responsibilities that businesses have to “protect, respect, remedy” human rights violations. Most transnational businesses are aware of these principles and in crafting recommendations on what the private sector could or should do on this issue, I would encourage the authors to become familiar with this guidance as well as other legal and policy instruments that are used in practice by both industry and the actors that seek to influence them.

Last, I fail to see how high seas closures would have a substantial effect on this issue – this recommendation feels like it was bolted on to the paper and has little potential as a real-world solution to these social issues in fisheries. A bigger effect would probably be generated by holding businesses to the Ruggie principles, and working with high-risk countries to implement domestic policies, backed by sufficient enforcement capacity, to uphold well-established standards of human rights that international law and policy has established (see Kittinger et al. 2017 Table S1). This would have the effect of reducing these human rights violations, essentially removing these subsidies, and curtailing much overfishing by making many fisheries uneconomic.

Below are additional specific comments that, if addressed together with these general comments, I believe would strengthen the paper.

Specific Comments:

- Line 60+: There are several definitions of modern slavery, but I would refer you to a good discussion of these by Siddharth Kara in his recently released book, *Modern Slavery*. In this he defines slavery as such: “Slavery is a system of dishonoring and degrading people through the violent coercion of their labor activity in conditions that dehumanize them.” (pg 8)
- Line 60+: Authors should also point out here (as they have in other parts of the article) that these abuses are not limited to the developing economies of the world. Indeed there have been well-documented cases in developed economies such as New Zealand (Marschke et al. 2016), Hawaii (Associated Press), Scotland, and other locations.
- Lines 92-101: I agree with the authors that there has been far too little attention to these issues among businesses in the seafood sector. However, a range of initiatives – some of which have been started and supported fully by businesses – have sprung up in the past several years (e.g., the Thai Shrimp Working Group – which was started by Costco, and involves major retailers such as Walmart). Additionally, while the majority of existing certification schemes such as MSC do not include social aspects, this is also changing. For example, FairTrade has launched a seafood program, which has certified several fisheries and is growing. Additionally the Monterey Bay Aquarium’s Seafood Watch program is developing a labor standard, Seafish’s Responsible Fishing Scheme is addressing these issues in the UK, the ILO and FAO are convening the Vigo Dialogues around these issues, and tools such as the Labor Safe Screen are being developed to support transparency on these issues. In aquaculture, ASC and GlobalGAP certifications include social elements. “Pre-certification” efforts such as Fishery Improvement Projects are also beginning to include social elements (e.g., see ASEAN FIP protocol). Many if not all of these efforts are not covered in the academic literature but in the practitioner community they are becoming more well-known. My advice is that the authors should endeavor to strike a balance here – clearly there is a major gap here in information, but there is also a flurry of activity and the joining of environmental and social issues represents an opportunity to think of sustainability in terms of both the fish and the people.
- 148-174: The majority of this focuses on the EU situation. I’d add a bit more about the US

situation as it is a similarly large market. Currently it only gets one mention in the last line.

- 181: Suggest you cite Kittinger et al. 2017

- 184: See my notes above about a wide range of initiatives in this space. The SeaBOS initiative is fantastic; it joins a set of initiatives in this space and a decade of investment by NGOs and foundations in buyer commitments and sourcing policy, which heretofore have been primarily focused on environmental issues and now – thanks to the media pressure – have begun to focus more fully on issues of social sustainability.

- 198-199: I understand that at the country level the fact that many of these nations have distant water fishing fleets may be a shared attribute. However, I think the authors should also mention that many of the most egregious documented cases of slavery and labor abuses have happened in fisheries within the EEZs of these countries – including in Thailand and Indonesia.

- 217-215: In other contexts this has been described as a sort of “displacement effect” - as societies become more affluent, they extract resources from ecosystems further afield (e.g., Cinner et al. 2009). The authors could tie in their theme on social justice here as this is essentially embedded in a global phenomenon of wealthier countries consuming the resources of the poor in less developed economies, as noted in lines 224-6.

- 258-260: this is an especially important point

- 296-304: I wonder if the authors can go further here. The use of VMS and AIS-based systems is great for tracking vessels (and SAR is being developed for “dark” vessel detection), but a whole range of technologies are being piloted to address some of these issues, including by groups like the International Labor Rights Forum. It would be great to highlight at least a few more examples to give readers a sense of the broad set of technologies that are emerging that offer promise in dealing with these issues.

- 305: There is also an emerging literature on traceability in the seafood sector that could be referenced here (e.g., Boyle 2012; Hardt et al. 2017; Lewis and Boyle 2017)

- 313-316: I believe the insurance industry is also starting to be more diligent on these issues, as it is a risk factor to them. (see article: <https://www.environmental-finance.com/content/news/insurers-black-list-illegal-fishing-boats.html>)

- 320: See the US’ new Seafood Import Monitoring Program. It is designed to thwart IUU products from being imported and will in essence require importers to have traceability on their products. I believe it was based in part on EU policy. See also these key policies for the US:

o US Trade Facilitation and Trade Enforcement Act <https://www.ap.org/explore/seafood-from-slaves/Obama-bans-US-imports-of-slave-produced-goods.html>

o California Transparency in Supply Chains Act: <https://oag.ca.gov/SB657>

References

Boyle, M. 2012. Without a Trace II: An updated summary of traceability efforts in the seafood industry. FishWise, Santa Cruz, CA. [online]

http://www.sustainablefishery.org/images/fishwise_traceability_writing_paper_august_2012.pdf.

Cinner, J. E., T. R. McClanahan, T. M. Daw, N. A. J. Graham, J. Maina, S. K. Wilson, and T. P. Hughes. 2009. Linking social and ecological systems to sustain coral reef fisheries. *Current Biology* 19:206-212.

Hardt, M. J., K. Flett, and C. J. Howell. 2017. Current Barriers to Large-scale Interoperability of Traceability Technology in the Seafood Sector. *Journal of Food Science* 82:A3-A12.

Kittinger, J. N., L. C. L. Teh, E. H. Allison, N. J. Bennett, L. B. Crowder, E. M. Finkbeiner, C. Hicks, C. G. Scarton, K. Nakamura, Y. Ota, J. Young, A. Alifano, A. Apel, A. Arbib, L. Bishop, M. Boyle, A. M. Cisneros-Montemayor, P. Hunter, E. Le Cornu, M. Levine, R. S. Jones, J. Z. Koehn, M. M. Redwood, J. G. Mason, F. Micheli, L. McClenachan, C. Opal, J. Peacey, S. H. Peckham, E. Schemmel, V. Solis-Rivera, W. Swartz, and T. A. Wilhelm. 2017. Committing to Socially Responsible Seafood. *Science* 356:912-913.

Lewis, S. G., and M. Boyle. 2017. The Expanding Role of Traceability in Seafood: Tools and Key Initiatives. *Journal of Food Science* 82:A13-A21.

Marschke, M., and P. Vandergeest. 2016a. Slavery scandals: Unpacking labour challenges and policy responses within the off-shore fisheries sector. *Marine Policy* 68:39-46.

Reviewer #3 (Remarks to the Author):

Review: Modern slavery facilitates overfishing in global fisheries

This paper tests and accepts the hypotheses that a link exists between modern slavery and fisheries governance and that slavery is most prevalent in poorly-regulated and cost-driven distant-water fleets. The study is novel and the subject will be of interest to readers in the wider fields, as many commercial fish species have been overexploited, and the modern slavery is a sad reality that should be stopped. However, I have a range of concerns from data quality to analytical method and from results to conclusion. Due to a lack of scientific rigor, perhaps the paper can be revised as a perspective article rather than a research article.

Data

The data used to test the hypothesis include: global fish catch by countries, estimated unreported catch, estimates of fisheries subsidies, distance of fishing activity from the home country capital city, global trade flows for seafood commodities, and prevalence of slavery based on interview.

The paper does not explain how "unreported" catch is estimated and how large is the uncertainty. Because catch and fishing activities are unreported, how do you know fishing locations and the distances from the capital city and how accurate are the estimates?

Modern slavery encompasses many industries and sectors. The paper lists textile, agriculture, construction and fisheries sectors, as well as in the sex industry and in forced marriages. It is unclear whether the total estimated slavery is used or only slavery in fisheries is used in establishing the hypothetical relationships. Because the Methods section doesn't explain this, I suspect that the total estimated slavery is used. If this is true, assuming that slavery in fisheries is the same as the total slavery may be too risky, and the results may not be meaningful. Further, the estimated 40 million people in modern slavery is based on 42 thousand interviews in 25 countries. The process implies that on average each interview results in about one thousand estimated slaves. The estimates may involve high uncertainty within the surveyed countries. The estimated number of slavery in unsurveyed countries by model extrapolation must be more uncertain.

The paper states that 45% to 50% of domestically consumed seafood is imported wild-caught fish in US and EU, and that it is not possible to distinguish between farmed and wild caught products. Hence, in addition to the uncertain numbers of slaveries in fisheries, the impact of imports on the slavery risk of domestically consumed seafood is questionable. Isn't it more likely that most of wild-caught fish comes from countries with low slavery prevalence because of their better managed fisheries, while most of farmed fish comes from countries with high slavery prevalence because of their high aquaculture production?

Method

Linear regression is used to test the relationships between slavery and (i) unreported catch, (ii) landed value, and (iii) unreported catch and landed value combined. Because both dependent and independent variables contain high uncertainty, these errors, either estimated or assumed, should have been considered in the model. Instead of combining the two independent variables, multiple regression is a more straightforward approach.

Data on modern slavery are available for 167 countries, but the study only uses 20. No clear reason is given why most countries are not included in the analysis. The reason for excluding India is its high land-based slavery. Is the decision based on data or speculation? If the land-based slavery data are available for India, why does the paper not include India by separating land-based slavery and slavery in fisheries?

The purpose of doing the principal components analysis is unclear; so is its outcome regarding slavery. The CPA separated 20 countries into three clusters based on several fisheries and economic variables but not slavery.

Results and conclusion

The results show a positive (but weak) correlation between slavery and unreported catch and between slavery and landed value. However, correlation does not imply causation. Slavery is a potential cause of unreported catch, but there are other more important factors causing unreported catch, including fisheries laws and regulations, government's ability to monitor and enforce the regulations, fisheries characteristics, the state of the economy in a country, etc. Similarly, these factors, as well as culture, can affect mean landed value (for example small fish is popular food in countries of poor performance (the red cluster)).

Other comments

The title is a bit too confident for a research paper.

A large part of the Discussion talks about how to resolve the problem of modern slavery, which is less relevant to other sections.

The list of references is too long. News from the internet and grey literature should be avoided.

Table S1 is not found in the paper.

Reviewer #4 (Remarks to the Author):

See the attached [document]

Reviewer #1:

Comment: This is a novel, important contribution. I appreciate your analysis, drawing on a series of big 'n' data sets that cover global data on fish catch, trade and modern slavery, to really highlight the challenges of working conditions in off-shore fishing vessels. Your visuals are compelling and easy to follow for a non-stats expert (me!). I applaud your timely work.

Response: We thank the reviewer for their encouragement, and hope that our revised manuscript can adequately address their remaining questions and concerns.

Comment: Framing: The concept of modern day slavery is appealing (and for men facing the worst working conditions this term is appropriate) and framing your paper in this way also makes for an explicit link to the Modern Slavery Index. However, you can go further than this, acknowledging that a range of poor working conditions exist within the fisheries (from human trafficking to practices that may be poor but that men agree to and negotiate). The idea of a continuum of unfree labour is helpful here. This is to also show that there are cases where men exert agency and to not only paint them as victims. It is also worth pointing out how little unpacking of labour abuse in fisheries exists, and this is where migration scholars have done some thoughtful analysis.

Response: This is an important point, and the fact that modern slavery exists only at one extreme of a broader spectrum of abusive labour conditions is one we have acknowledged explicitly in the introduction (lines 97-102). Specifically, we make clear that substandard working conditions can be entered into voluntarily in some instances and are often legal in the jurisdiction in which they occur. Further, we also note that inconsistency in labour regulations in part explains the appeal to vessel owners of “flag of convenience” registries, which allow for significantly relaxed safety and employment rules and conditions² (lines 84-87). However, for the analysis, we have retained the focus on modern slavery as quantified in the Global Slavery Index, as this is our source of data.

Comment: Discussion and conclusions. I would like to see you go further here. For example, if laws are to be enforced perhaps part of this is to incite that ILO C-188 is ratified. I also think you can add more detail on the Thai case as NGOs, IOs and the business community also play a strong role in the significant progress (to show it is not only New Zealand but also global South countries who are capable of thinking this through – Vandergeest et al. 2017 provide a brief overview of how Thai NGOs have been involved, and I agree that we don't know if this will be sustainable given it is a Military government).

Response: We thank the reviewer for this comment. We have further emphasised the importance of ILO C-188 including in the case of Thailand, also noting that ratification to date is limited to only 10 countries, which also requires addressing (lines 416-427). We have expanded the discussion of existing slavery cases and responses in both the Introduction and Discussion (e.g. lines 438-440, 453-456). We also note the Issara Institute's recent report on attitudes among the Thai fishing industry (i.e. employers) where respondents demonstrated both a lack of awareness that activities such as the withholding of wages or the imposition of recruitment expense debts onto crew even constituted forms of labour abuse, and scepticism of and a degree of resentment to the new measures being implemented by the Thai government. We have drawn attention to this in the revised manuscript in the context of the need for both education within the industry, and also the need for trading partners to support governments in places like Thailand in order to ensure

that new systems are effective and do not lose legitimacy through poor implementation (lines 443-452)

Comment: Also, in terms of supply chain accountability, perhaps add some more detail in. For example, certification generally has focused on global North fisheries and even here there are ecological issues. Even if MSC was to add in labour issues (which they are not in doing in any significant way), they are not really certifying global South fisheries anyways. How do companies insist on supply chain accountability and traceability given that serious labour abuse occurs in the global South (but not only)? While companies such as Thai Union have all seafood operations go in-house, in a sense creating sustainability enclaves for global North consumption, how do we ensure basic human rights for fish labourers in addition to insisting on more sustainable fishing practices? Is FairTrade Tuna a model to follow? <http://3blmedia.com/News/Safeway-and-Fair-Trade-USA-Launch-Worlds-First-Fair-Trade-Certifiedtm-Seafood>. I realize there are not any easy answers here, but more to see you probe a bit more as your writing team has considerable depth on these issues.

Response: We thank the reviewer for this very valuable comment. We have now provided much more detail in the revised manuscript where we discuss the current certification schemes and the emerging developments (such as Monterey Bay Aquarium/Liberty Asia's Seafood Slavery Risk Tool). See lines 163-183

Comment: Minor detail: Refs 29-33 really refer to Burmese men working in Thailand, Philippine men trafficked across a number of industries, and Burmese and Khmer men on Thai boats in Indonesian waters etc. Little has been flagged in terms of Burmese fisheries, for example. You can easily update / add nuance to this section, drawing on more recent reports, to signal labour abuse scandals emerging in Thailand, Taiwan (the Greenpeace 'Turn the Tide' report looks at labour abuse in Taiwanese fisheries), Ireland or Hawaii just in 2014-2016 while Christina Stringer's work in New Zealand is helpful for understanding the role of Indonesian workers on Korean vessels in the early 2010s.

Response: Our section discussing existing slavery cases has been expanded in line with the reviewer's suggestions, including the latest allegations of slavery in the UK scallop fishery, which has been one of the first products identified in the Seafood Slavery Risk Tool mentioned above (lines 111-126)

Comment: Other papers to add to your analysis:

- Lewis & Boyle, 2017. The Expanding Role of Traceability in Seafood. *Journal of Food Science*.
- Kittinger et al. 2017 Committing to Socially Responsible Seafood. *Science*.

Response: We do cite Kittinger et al. (2017) in the original manuscript, and now make repeated reference to this important paper, setting our study in the context of their call for greater attention to social issues associated with fisheries (line 189). The Lewis and Boyle paper is a valuable addition to our overview of traceability issues (lines 511-585), and we thank the reviewer for pointing us to this contribution.

Reviewer #2 (Remarks to the Author):

Comment: The authors are to be commended for undertaking an ambitious analysis assessing the relationship between overfishing and modern slavery. The study provides an important contribution to our understanding of this important issue and I believe it is worthy of acceptance, following some minor revisions.

Response: The reviewer's comment is much appreciated and we hope that the revised manuscript satisfactorily addresses your remaining concerns.

Comment: First, the authors make the point that our global understanding of the scale of this issue in the seafood sector is quite limited. The Global Slavery Index is one of the only global level indicators available, but it is not focused specifically on the seafood industry and its survey-based methodology has some limitations (as noted in line 228+). I think the relationship the authors have found certainly highlights the co-occurrence of these issues, but without stronger evidence I don't believe at this point you can say definitively that slavery facilitates overfishing. The authors present a good set of arguments and lines of reasoning for how this relationship can work, but without definitive analysis and evidence of the subsidy effect I think at this point you are pointing out two main (and still very important) issues:

1. Overfishing and modern slavery co-occur in areas with poor governance which likely mediates both and there is strong potential for poor labour practices to provide subsidy effect for fisheries
2. Products from these geographies are flowing into markets in the EU and US which has serious implications for trade as well as for the solutions required to address these issues

If the authors are still very keen on making the link on the subsidy effect, one suggestion is to assess how many fisheries would become uneconomic if labour practices were just and human rights were fully protected. How much of the overfishing problem would this fix?

Response: We thank the reviewer for this valuable comment. Given the high-level nature of our analysis, as also recognised by the reviewer, we feel that attempting to quantify the subsidy effect is beyond the scope of the current study and available data. It does however identify an important avenue for future data collection and research, and we make this point clear in concluding our revised manuscript (lines 643-647). We have also tempered our language on this point, limiting ourselves to mentioning it in the context of the documented contribution of labour costs to overall costs of fishing (lines 44-53). Labour costs represent a substantial component of fishing costs globally, and over 40% in Asia where many of the most serious cases of modern slavery have occurred, a potential strong incentive to employ cheaper labour through informal or illegal channels. We also note if cost reduction/profit maximisation is not an objective of slave use on fishing boats, it is not entirely clear what motive would generate this behaviour. We have more clearly expressed this in the revised manuscript.

Comment: Second, it is important to note at the outset that slavery and associated human rights and labour violations are among the most egregious practices globally in the seafood sector. However, they are not the only issues. Social sustainability can mean a lot of different things to different groups. A recent effort (Kittinger et al. 2017) focused on

generating a global consensus definition of social responsibility for the seafood sector. This initiative took a similar approach as was taken with environmental sustainability parameters, which derived primarily from the FAO's Code of Conduct. Similarly, the social responsibility framework derived primarily from the FAO's Voluntary Guidelines for Small-scale fisheries, which like the code of conduct was developed globally with significant stakeholder input. I recommend the authors at a minimum provide some context on these other issues, while keeping their analytical focus on the analysis of human rights and slavery.

Response: We agree that Kittinger et al.'s paper is an important call for marine and fisheries practitioners to pay greater attention to the social issues that run in parallel with ecological threats. We have acknowledged that the situations of modern slavery on which our paper focuses lie at one extreme of a continuum of abuse (a point also raised by the first reviewer), while retaining our focus on the more clearly defined slavery cases (lines 97-102). We have and also set our work in the context of Kittinger et al.'s call for more interdisciplinary work in this space (line 189).

Comment: Third, the authors' risk-based approach is a good one but it can be problematic at the country level. For a country like Indonesia, for example – where known labour violations and slavery have occurred – this means that a country-level rating obscures a wide range of performance from good to egregious. This has the effect of painting all the fisheries in a complex geography with a single risk rating, which is problematic both for well-performing fisheries (socially and environmentally) who may not be rewarded in the marketplace, as well as the most poorly performing fisheries, which may be actually, require immediate attention, and may be much worse than a single risk rating could convey. Also – the risk-based analysis at the country level can incorrectly convey that this is a problem that needs to be addressed via domestic policy in each of these geographies. However, due to the high degree of migrant labour in fleets, actions almost certainly require better policy and enforcement within the country in which the violation occurs (or in the case of DWFN, on vessels flagged in these countries) but also the source country for the labourers.

Response: We appreciate the reviewer for stressing the potential shortfalls of country-level assessments. We consider the work we present in our manuscript as falling into the category of first-order, country-level assessment, as is also done by other global assessments, such as the Environmental Performance Index and the Ocean Health Index. We recognize that such country-level assessments do have their shortfalls, and we acknowledge this in the revised manuscript. Our analysis is certainly not intended to mask within-country variability – clearly there are pockets of best practice in even the most challenged areas, as shown by the Fair Trade certification awarded to a small group of Indonesian tuna fisheries (lines 350-357). However, the key thesis of our paper is that there are structural elements, rooted in over 50 years of well-intentioned but misguided development of industrial fisheries, that have created systemic problems in fisheries. If some parts of the industry are continuing to remain profitable by fishing legally, sustainably and ethically, then this is certainly to be celebrated, but the evidence for widespread problems in fisheries is increasing and there is a need to identify risk. Addressing these issues transcends national borders and requires the involvement of fishing nations, labour supply countries, and consumer markets as all have roles and responsibilities in the process. These are points that we hopefully make clearly throughout the revised manuscript.

Comment: Fourth, I recommend the authors acquaint themselves with the UN Guiding Principles of Business and Human Rights (also known as the “Ruggie principles”; http://www.ohchr.org/Documents/Publications/GuidingPrinciplesBusinessHR_EN.pdf). This well-known guidance document describes the clear corporate responsibilities that businesses have to “protect, respect, remedy” human rights violations. Most transnational businesses are aware of these principles and in crafting recommendations on what the private sector could or should do on this issue, I would encourage the authors to become familiar with this guidance as well as other legal and policy instruments that are used in practice by both industry and the actors that seek to influence them.

Response: This was a valuable reference, and we have integrated this into the section where we discuss industry responsibilities to social values in their businesses (lines 465-483). Clearly, even non-binding guiding principles can have a powerful effect on socially aware businesses. However, business motivation to pursue improvements in their supply chains in the absence of the threat of legal sanction or financially damaging reputation loss may not always be strong. There thus remains a need for governments and consumers to provide structural change and other ‘encouragement’. We have emphasized this in the revised manuscript.

Comment: Last, I fail to see how high seas closures would have a substantial effect on this issue – this recommendation feels like it was bolted on to the paper and has little potential as a real-world solution to these social issues in fisheries. A bigger effect would probably be generated by holding businesses to the Ruggie principles, and working with high-risk countries to implement domestic policies, backed by sufficient enforcement capacity, to uphold well-established standards of human rights that international law and policy has established (see Kittinger et al. 2017 Table S1). This would have the effect of reducing these human rights violations, essentially removing these subsidies, and curtailing much overfishing by making many fisheries uneconomic.

Response: We have expanded our discussion on the Ruggie principles as discussed in the previous point.

With respect to the role of marine protected areas (MPAs) and (partial) high seas closures, data on reconstructed fisheries show clear declines in global catches, with “peak fish” having occurred in 1996 (Pauly and Zeller 2016), with implications for both food security and profitability. Decades of research show that MPAs support fisheries sustainability, and fisheries economists have argued convincingly that even a full closure of the high seas would have only a slight short term impact on fish catches while greatly improving equity in the distribution of fisheries returns among countries. We do not suggest that MPAs and high seas closures are a silver bullet but rather that they are an important complementary tool to the supply chain and enforcement and monitoring measures discussed in the paper. Conventional fisheries management has evidentially been unable to halt fisheries declines nor to ensure social justice in the labour force. As we demonstrate a correlation between labour abuse and distant water fishing fleets, consideration of more effective management of the high seas is critical. Advances in vessel monitoring technologies are also assisting (e.g. AIS: see Kroodsma et al., *Science* 359, 904–908 (2018)), but are still reliant on their adoption and use by fishers. We have therefore retained the suggestion of improved high seas management (including high seas MPAs and other closures), as it has genuine merit

as evidenced by the larger MPA and fisheries economics literature, but have softened our approach and better integrated it into our suite of recommendations (lines 587-610).

Comment: - Line 60+: There are several definitions of modern slavery, but I would refer you to a good discussion of these by Siddharth Kara in his recently released book, Modern Slavery. In this he defines slavery as such: “Slavery is a system of dishonoring and degrading people through the violent coercion of their labour activity in conditions that dehumanize them.” (pg 8)

Response: Thank you for pointing us to this interesting source. There are indeed a number of definitions of modern slavery. The definition we use is derived from the International Labour Organisation and mirrors that used by the Global Slavery Index, which is the data we use. It is both widely accepted and also, we feel, has utility in its use of language that clearly defines specific actions on the part of employers as constituting modern slavery, such as the withholding of wages. As such we have retained our existing definition and sources.

Comment: - Line 60+: Authors should also point out here (as they have in other parts of the article) that these abuses are not limited to the developing economies of the world. Indeed there have been well-documented cases in developed economies such as New Zealand (Marschke et al. 2016), Hawaii (Associated Press), Scotland, and other locations.

Response: We thank the reviewer for pointing out our oversight at this point in the manuscript. The developed world cases are now included in this section and we have striven to provide a balanced overview of the known slavery cases (lines 111-126).

Comment: - Lines 92-101: I agree with the authors that there has been far too little attention to these issues among businesses in the seafood sector. However, a range of initiatives – some of which have been started and supported fully by businesses – have sprung up in the past several years (e.g., the Thai Shrimp Working Group – which was started by Costco, and involves major retailers such as Walmart).

Response: We acknowledge the work done by retailers and their business partners to address labour issues, but note that much of this has been purely reactive in the face of reputational risk following media exposure (lines 156-162). This contrasts with the ambitions of the Ruggie Principles, for example. Whilst it is important that companies are moving to address this issue, it has only become necessary for them to do so thanks to the activities of the Associated Press and the NGOs who brought the abuses in their supply chains to light. We feel that legislation, in particular, but also increased consumer demand are needed to encourage companies to move beyond ‘fire-fighting’ to a more proactive stance on social and environmental sustainability. In this context we recognise in the text the recently initiated Seafood Business for Ocean Stewardship (SeaBOS) initiative (line 152-156).

Comment: Additionally, while the majority of existing certification schemes such as MSC do not include social aspects, this is also changing. For example, FairTrade has launched a seafood program, which has certified several fisheries and is growing.

Additionally, the Monterey Bay Aquarium's Seafood Watch program is developing a labour standard, Seafish's Responsible Fishing Scheme is addressing these issues in the UK, the ILO and FAO are convening the Vigo Dialogues around these issues, and tools such as the Labour Safe Screen are being developed to support transparency on these issues. In aquaculture, ASC and GlobalGAP certifications include social elements. "Pre-certification" efforts such as Fishery Improvement Projects are also beginning to include social elements (e.g., see ASEAN FIP protocol). Many if not all of these efforts are not covered in the academic literature but in the practitioner community they are becoming more well-known. My advice is that the authors should endeavour to strike a balance here – clearly there is a major gap here in information, but there is also a flurry of activity and the joining of environmental and social issues represents an opportunity to think of sustainability in terms of both the fish and the people.

Response: We thank the reviewer for pointing out this gap in our manuscript. We have revised the manuscript to achieve a better balance, noting both indicative wins being achieved in this area while recognising that there are limits to their scope and coverage (line 182-185, 465-483). However, it is also not our intention to duplicate the work of others by attempting an exhaustive audit of the growing suite of individual government, NGO and industry responses in this paper with, for instance, Lewis and Boyle (2017) providing a dedicated review of supply chain initiatives. Rather our goal has been to take a step back from the media exposés and the reactions they have produced in the sector, and to ask whether there are fundamental drivers in industrial fisheries that increase the risk of modern slavery and whether an understanding of these can help drive policy at a broader level that addresses root causes.

Comment: - 148-174: The majority of this focuses on the EU situation. I'd add a bit more about the US situation as it is a similarly large market. Currently it only gets one mention in the last line.

Response: We have rewritten the section to place greater emphasis on both regions (lines 285-301).

Comment: - 181: Suggest you cite Kittinger et al. 2017

Response: Included (now line 189)

Comment: - 184: See my notes above about a wide range of initiatives in this space. The SeaBOS initiative is fantastic; it joins a set of initiatives in this space and a decade of investment by NGOs and foundations in buyer commitments and sourcing policy, which heretofore have been primarily focused on environmental issues and now – thanks to the media pressure – have begun to focus more fully on issues of social sustainability.

Response: Please see our response to the comment above; we have made additional reference to this initiative (lines 152, 313, 478)

Comment: - 198-199: I understand that at the country level the fact that many of these nations have distant water fishing fleets may be a shared attribute. However, I think the

authors should also mention that many of the most egregious documented cases of slavery and labour abuses have happened in fisheries within the EEZs of these countries – including in Thailand and Indonesia.

Response: We thank the reviewer for emphasising that even domestic fleets may have labour abuses and modern slavery problems. We have emphasized this in the revised manuscript. For example, Thailand does appear to have significant domestic issues in terms of poor oversight applied to fisheries. In our multivariate (PCA) analysis Thailand has a higher value for unreported fishing and a lower value for distance than, e.g., Taiwan and South Korea, so it may be that their lack of governance is a major driver. However the portion of the Thai fleet that operates outside their EEZ, albeit in the waters of neighbouring countries, is clearly also a serious issue so we feel that ‘distant-water operations’ remain a factor worth highlighting in our analysis. This has been commented on in the results section (lines 253-257).

Comment: - 217-215: In other contexts, this has been described as a sort of “displacement effect” - as societies become more affluent, they extract resources from ecosystems further afield (e.g., Cinner et al. 2009). The authors could tie in their theme on social justice here as this is essentially embedded in a global phenomenon of wealthier countries consuming the resources of the poor in less developed economies, as noted in lines 224-6.

Response: We thank the reviewer for the Cinner et al. reference, which provided some valuable theoretical background to the point we were trying to make. We have now observed in our revised manuscript that globalised industries in all sectors have routinely outsourced environmental and social problems of production to other, less regulated parts of the world, and have taken this opportunity to reference the ‘scale effect’ discussed in Cinner et al. to further emphasize this point (lines 361-375).

Comment: - 296-304: I wonder if the authors can go further here. The use of VMS and AIS-based systems is great for tracking vessels (and SAR is being developed for “dark” vessel detection), but a whole range of technologies are being piloted to address some of these issues, including by groups like the International Labour Rights Forum. It would be great to highlight at least a few more examples to give readers a sense of the broad set of technologies that are emerging that offer promise in dealing with these issues.

Response: We have elaborated on these technologies but an exhaustive list would detract from the core thesis. Moreover, as other researchers (e.g. Lewis and Boyle, 2017) have done a thorough review of traceability techniques and technology, we have more clearly directed readers to their work while seeking to provide an overview in our text (lines 561-588).

Comment: - 305: There is also an emerging literature on traceability in the seafood sector that could be referenced here (e.g., Boyle 2012; Hardt et al. 2017; Lewis and Boyle 2017)

Response: We thank the reviewer for these suggestions, which we have incorporated into the section on traceability (lines 514-588).

Comment: - 313-316: I believe the insurance industry is also starting to be more diligent on these issues, as it is a risk factor to them. (see article: <https://www.environmental-finance.com/content/news/insurers-blacklist-illegal-fishing-boats.html>)

Response: This is an important point which we have incorporated into the legal response section in the context of effectively removing a vessel's license to operate by making the financial risk of operating uninsured too great (lines 507-513).

Comment: - 320: See the US' new Seafood Import Monitoring Program. It is designed to thwart IUU products from being imported and will in essence require importers to have traceability on their products. I believe it was based in part on EU policy. See also these key policies for the US:

US Trade Facilitation and Trade Enforcement Act <https://www.ap.org/explore/seafood-from-slaves/Obama-bans-US-imports-of-slave-produced-goods.html>

California Transparency in Supply Chains Act: <https://oag.ca.gov/SB657>

Response: We thank the reviewer for drawing our attention to these measures and have incorporated them into our overview of supply chain legislation, in addition to the recent measures on corporate due diligence adopted by France (lines 531-559).

Reviewer #3 (Remarks to the Author):

Comment: This paper tests and accepts the hypotheses that a link exists between modern slavery and fisheries governance and that slavery is most prevalent in poorly-regulated and cost-driven distant-water fleets. The study is novel and the subject will be of interest to readers in the wider fields, as many commercial fish species have been overexploited, and the modern slavery is a sad reality that should be stopped. However, I have a range of concerns from data quality to analytical method and from results to conclusion. Due to a lack of scientific rigor, perhaps the paper can be revised as a perspective article rather than a research article.

Response: We appreciate the reviewer's enthusiasm for the topic and agree that this is a topic that requires increased attention, hence our motivation to undertake the analysis. Please see below where we respond to the reviewer's specific comments on rigor and analysis. The article is however best placed as a research rather than perspective article given the significant analysis of existing global datasets.

Comment: The data used to test the hypothesis include: global fish catch by countries, estimated unreported catch, estimates of fisheries subsidies, distance of fishing activity from the home country capital city, global trade flows for seafood commodities, and prevalence of slavery based on interview.

The paper does not explain how "unreported" catch is estimated and how large is the uncertainty. Because catch and fishing activities are unreported, how do you know fishing locations and the distances from the capital city and how accurate are the estimates?

Response: Given the well-established nature of the reconstructed and spatially allocated catch data, we consider it sufficient to reference the relevant literature on this topic (Palomares et al. 2016. Still catching attention: Sea Around Us reconstructed catch data, their spatial expression and public accessibility. *Marine Policy* 70: 145-152). Specifically, the majority of the fisheries related data were obtained from the work of the Sea Around Us project, which has researched and documented estimates of unreported catches for every country in the world as part of the global catch reconstruction project. The project spanned over 10 years and involved more than 300 international collaborators. The methodology is described in detail in Zeller et al. 2016, *Marine Policy* 70: 145-162 and the global findings were published in Pauly & Zeller 2016, *Nature Communications* 7: 10244. Uncertainty has also been addressed in Pauly & Zeller 2017, *Marine Policy* 81: 406-410. Reconstructed data for well over 120 individual countries have been published in the peer-reviewed literature and reconstructed data can be considered the best available data for total catches, as they complement officially reported data with comprehensive and detailed yet conservative time series estimates of unreported catches. Equally, the spatial allocation of fisheries catches (where catches have been taken) has been thoroughly vetted in the scientific literature over the last 15 years and is well described. All of these reference are appropriately cited in the methods section.

Comment: Modern slavery encompasses many industries and sectors. The paper lists textile, agriculture, construction and fisheries sectors, as well as in the sex industry and in forced marriages. It is unclear whether the total estimated slavery is used or only slavery in

fisheries is used in establishing the hypothetical relationships. Because the Methods section doesn't explain this, I suspect that the total estimated slavery is used. If this is true, assuming that slavery in fisheries is the same as the total slavery may be too risky, and the results may not be meaningful. Further, the estimated 40 million people in modern slavery is based on 42 thousand interviews in 25 countries. The process implies that on average each interview results in about one thousand estimated slaves. The estimates may involve high uncertainty within the surveyed countries. The estimated number of slavery in unsurveyed countries by model extrapolation must be more uncertain.

Response: We specifically noted in the original manuscript that we were using the national prevalence estimate of slavery per country, which includes all sectors. There are no global fisheries-specific estimates as yet and indeed, one of the reasons we proceeded with this analysis was to raise the issue so that improved sector-specific estimates are seen as a priority and thus obtained. However, we have further clarified that the global slavery estimates are cross-sectoral (e.g. lines 325-336, and methods).

The value in this analysis lies in highlighting systemic drivers that place labour at risk in the fisheries sector. We feel that it is instructive to examine the correlation between countries' performance in fisheries and their domestic track record on labour rights issues at the global level noting that our general results are consistent with anecdotal reports of slavery in the fisheries sector. With respect to uncertainty, the GSI explicitly documents this with the full data set and methodology available for review. The primary data source behind this edition of the GSI, the Gallup surveys, has since been used as the basis of the more recent ILO-WFF Global Estimates of Modern Slavery, suggesting a high degree of confidence by international organisations in this methodology.

Comment: The paper states that 45% to 50% of domestically consumed seafood is imported wild-caught fish in US and EU, and that it is not possible to distinguish between farmed and wild caught products. Hence, in addition to the uncertain numbers of slaveries in fisheries, the impact of imports on the slavery risk of domestically consumed seafood is questionable. Isn't it more likely that most of wild-caught fish comes from countries with low slavery prevalence because of their better managed fisheries, while most of farmed fish comes from countries with high slavery prevalence because of their high aquaculture production?

Response: Fundamental to this concern is whether slavery prevalence varies sufficiently in farmed and wild-caught products to affect the degree to which slavery-derived products are present in the US and EU markets. There are currently no sector-specific estimates of rates but it is important to recognise that there are well documented examples of slavery in both the aquaculture and wild-capture sectors.

The second point rests on the assumption that most seafood imports to the US and EU are from fisheries with good practice. This is clearly not the case. For instance, Alder & Sumaila (2004) in their paper: Western Africa: the fish basket of Europe past and present in the *Journal of Environment and Development* 13(2): 156-178 have documented the flow of seafood into the EU. A further complicating factor is that the resolution of most trade data (e.g., "frozen fish fillets") make it almost impossible to differentiate either the actual species or product origin (wild caught versus farmed). We note also that one of the major criticisms of the Marine Stewardship Council certification programme is that it focuses largely on

developed country fisheries and thus gives a skewed picture of fisheries sustainability which means that fish products from developing countries are not generally assessed.

In terms of the trade analysis we present in our paper, we feel that we have been consistent in our approach by using metrics for seafood trade that necessarily imply aggregation across multiple processes (fishing, processing, transport, etc.), and a correspondingly broad metric for slavery at the national level (lines 288-291). In all likelihood, based on the limited data available from fisheries specific surveys on labour abuses, the *per capita* rate of modern slavery or labour abuses in fisheries would be higher than the national average in major fishing countries such as Thailand. As such, our estimates may likely be biased downwards. The purpose of the trade analysis is to demonstrate, using robust and validated data, that the reliance of developed countries on seafood imports from countries with currently poor records on addressing modern slavery overall must, all other things being equal, necessarily increase their risk of consuming slave-derived seafood. This hopefully makes clear the need and, one might argue, the responsibility for wealthy consumer countries to partner with their 'suppliers' to improve net-to-table traceability and eliminate slavery at the production end. The involvement of the EU and US in assisting Thailand to improve its fisheries is a concrete and encouraging example of this. We have expanded on this point and also included additional material on trade and business involvement as, e.g., suggested by reviewer #2.

Comment: Linear regression is used to test the relationships between slavery and (i) unreported catch, (ii) landed value, and (iii) unreported catch and landed value combined. Because both dependent and independent variables contain high uncertainty, these errors, either estimated or assumed, should have been considered in the model. Instead of combining the two independent variables, multiple regression is a more straightforward approach.

Response: We did in fact use multiple regression to determine the combined influence of both of our independent variables on slavery prevalence (Fig. 1f). We also included the single variable regressions (Figs 1d and 1e) to illustrate the relationships between slavery and each of the two independent variables that are subsequently used in the multivariate regression (Fig 1f).

We thank the reviewer for the question on estimation errors, as it motivated us to perform a sensitivity analysis (now included as supplementary information) using a range of estimate values for both fisheries inputs (based on the spread of annual values used to derive the 10-year averages used) and the slavery prevalence estimates (based on the confidence intervals of the survey estimates). We continue to believe that the published (and vetted) slavery estimates of the GSI, as well as the long-run average catch and price data for fisheries, are the appropriate metrics to use. Furthermore, our confidence in our reported results is bolstered by the fact that the model R^2 value falls within the 95% confidence range of 10,000 Monte Carlo simulations of the regression model based on inputs drawn from the probability distribution of all possible values for both dependent and independent variables (Methods section, Fig S2).

Comment: Data on modern slavery are available for 167 countries, but the study only uses 20. No clear reason is given why most countries are not included in the analysis. The reason for excluding India is its high land-based slavery. Is the decision based on data or

speculation? If the land-based slavery data are available for India, why does the paper not include India by separating land-based slavery and slavery in fisheries?

Response: The 20 countries were chosen based on their contribution to global fisheries – they represent 80% of global landings, making them the fisheries of greatest concern in identifying and tackling slavery in fishing. The decision to exclude India from the linear regression analysis was based on known details of the GSI data for India, which were collected at the state level to allow modern slavery levels in landlocked states to be used to infer the influence of land-based slavery on the whole-country estimate. As with the other country estimates, fishing and land-based slavery could not be explicitly disaggregated. Given the high prevalence of land-locked modern slavery in India, our approach was to treat India as an outlier for the linear regression analysis, until such time when data disaggregation of fishing and land-based slavery becomes feasible. This has been clarified in the methods section.

Comment: The purpose of doing the principal components analysis is unclear; so is its outcome regarding slavery. The PCA separated 20 countries into three clusters based on several fisheries and economic variables but not slavery.

Response: We thank the reviewer for drawing our attention to an important area requiring clarification in our methodology. The PCA and cluster analysis is designed to build on the results of the linear regression analysis. Having identified a relationship between fisheries attributes and the prevalence of modern slavery, we used information from a broader suite of country-level fisheries and economic metrics to determine what characteristics were shared between countries with a known history of serious labour abuses in fisheries. The objective of the analysis was to identify the suite of fisheries related factors that best explained known patterns of modern slavery in fisheries specifically and to identify other ‘at risk’ countries. GSI slavery estimates were not used in the final analysis since, as demonstrated by the linear analysis, they are themselves correlated with other fisheries and economic measures and thus were redundant. Prompted by your question we have clarified our rationale in the methods section of the manuscript.

Comment: The results show a positive (but weak) correlation between slavery and unreported catch and between slavery and landed value. However, correlation does not imply causation. Slavery is a potential cause of unreported catch, but there are other more important factors causing unreported catch, including fisheries laws and regulations, government’s ability to monitor and enforce the regulations, fisheries characteristics, the state of the economy in a country, etc. Similarly, these factors, as well as culture, can affect mean landed value (for example small fish is popular food in countries of poor performance (the red cluster)).

Response: We agree that there are a number of contributing factors to the magnitude of unreported fishing but it is important to note that we are not identifying correlates of unreported fishing. Rather we are asking whether unreported fishing explains part of the variation in the prevalence of slavery (our dependent variable in the regression). We agree that correlation is not causation but regression is a recognised tool for prediction. Having identified key predictors of slavery prevalence, we have a framework for assessing risk. To this end, the main point of the single and multiple regressions is to illustrate the broad

correlative relationship between slavery and headline measures of fisheries governance and profitability, to show that pockets of poorer performance on both sets of metrics co-occur, and to make the point that measures targeting fisheries sustainability issues and labour problems can usefully be undertaken together (and indeed will likely be addressing the same root cause). We have expanded and clarified our explanation of this issue in the revised manuscript (lines 220-230).

Comment: The title is a bit too confident for a research paper.

Response: We acknowledge the point of the reviewer, and have revised the title to better capture the main thesis of the paper.

Comment: A large part of the Discussion talks about how to resolve the problem of modern slavery, which is less relevant to other sections.

Response: We thank the reviewer for this comment. The discussion has been revised to better highlight the goal of the paper to demonstrate links between modern slavery and the current structural flaws in the management and operation of industrial fisheries. We agree with Kittinger et al. that our collective focus needs to shift from addressing fisheries ecological and social performance issues in isolation to seeing them as different manifestations of the same underlying problems. We therefore also feel it is important to include recommendations that address slavery specifically (such as the adoption of labour standards) as the connection between slavery and overfishing made here suggests that slavery *at least in part* provides a 'subsidy' to labour costs that allows some fisheries to persist in bio-economically unsustainable practices.

Comment: The list of references is too long. News from the internet and grey literature should be avoided.

Response: We have used peer-reviewed literature where available but it is important to note that much of the information on incidents of, and responses to, slavery in fisheries only exists in the media and grey literature. We have simplified URLs where possible, and hope that this approach is more reader friendly. The list of references has been purged of web-links, but 'grey literature' references have been retained, as they are important and well-cited NGO reports. We also consider it important to acknowledge the breadth and depth of work already done in this area in the NGO sector.

Comment: Table S1 is not found in the paper.

Response: We thank the reviewer for detecting this missing table. We apologize, this was our oversight during the initial manuscript submission, and has been corrected.

Reviewer #4

Comment: Overall I think the article is very interesting and discusses a topic that is very much in need of explicit discussion and evaluation. I think the authors did a great job at

discussing and covering all different elements of the topic. The article is well written and clearly laid out and I recommend publishing. My only (and main) concern with the paper is that I felt the analysis of the data was somewhat 'opaque' (and I'm not sure if that is the right word).

Response: We thank the reviewer for their positive feedback. We have substantially clarified the data analysis and hope this addresses this concern.

Comment: I know the method is pretty simple from a data analysis perspective but I would have preferred a separate methods section with a clearer outline of the steps in the data analysis process. First we established relationship between variable A and B and then looked at ... etc. A list of datasets used, the nature of the data, and their source would work well for me too! In my mind (and I may have the wrong end of the stick... but) there were four aspects to the data analysis 1) relationship between a number of global variables 2) a model to predict slavery, 3) a PCA typifying the types of drivers that may underlie/pin slavery 4) a product flow analysis linking where product came from and its destination. Each one of these needs more detail and the connection between them made needs to be more explicit.

Response: We thank the reviewer for this suggestion. We have substantially edited the methods section (at the end of the paper, after the figures) to take the reviewer's comment into account. We have also made the links between the analyses clearer and more explicit in the results section.

Comment: There are a few ambiguous statements on Page 4. High-risk environs to relatively lower risk market – please specify risk of what – slavery. Financial performance – financial performance of what – the fishing vessel – the sector as a whole?

Response: We appreciate this feedback. We have edited substantially for clarity and consistency throughout (e.g. lines 194, 277).

Comment: I feel a separate methods section is required (as mentioned above). A bit more detail is needed on for instance, tests of difference (what sort) and p values. In addition, a solid argument why the authors think that, for instance, a R2 value of 0.24 is high enough to make deterministic statements about the relationships between slavery, the presence of illegal fishing and lack of governance. The whole argument hinges on these relationship between the global data sets and I feel it could/should be a bit more convincing.

Response: We wonder if the journal-specific location of the Methods (at the very end of the paper) resulted in some confusion for reviewers as to where to look for this information. However, we have also substantially edited the section for the resubmission to take a number of requests for increased details and clarity into account.

Comment: Also – the link between two seemingly separate analyses 1) unreported catch, mean landed catch value, and prevalence of slavery and 2) the PCA should be better explained. If there is a logical sequence in doing these two analyses – how was that achieved. What can be better understood from doing these two analyses (i.e. the combined understanding you get which you wouldn't from only doing one of these).

Response: We thank the reviewer for raising this point and it was raised by other reviewers. We have substantially edited the methods and results section to clarify our rationale and the relationship between the various components. Briefly, the first set of linear models serves to demonstrate the co-occurrence of labour and fisheries performance issues in the major fishing countries of the world; the second analysis (PCA) identifies the factors that best explain the distribution of specific incidents of slavery on fishing vessels among these countries, helping us develop a more refined a risk model that indicates where additional slavery issues in fisheries may lie. (lines 198-209)

Comment: How was the model (figure 1 f) detailed? I cannot find this in the text. What sort of model? Was the model generated using all country data or only the top 20. Again, a clearly outline methods section might be able to address these question relatively easily.

Response: This figure presents the results of our multiple regression, demonstrating the relationship between observed and predicted prevalence of slavery. This has been clarified in the revised methods section. (lines 220-222, Methods section)

Comment: Some of the standard statistics that are generally reported with PCAs are missing (cumulative explained variance, Eigenvalues etc. PCA score by group for the different dimensions). Why three groups on two dimensions Etc.

Response: The choice of the PCA and clustering approach, and the tools used to estimate the optimum number of clusters, are detailed in the methods section of the paper. The PCA was used to simplify the data set to two principal components, and a cluster fitting heuristic was used to determine the optimum number of groups to adequately capture the variation in the resulting composite variables (PC scores) – these are the three groups. We have included some key diagnostic plots from the PCA as supplementary information (variance explained by the principal components and the contribution of each variable to variance explained, Fig S1).

Comment: Page 4 - Can the authors provide a rationale for choosing the top 20 volume countries? How much of total production/consumption trade etc applies to these countries (i.e. why can we ignore the remaining countries for analysis).

Response: We chose the top 20 as they are responsible for 80% of the world's industrial fishing landings. Thus, an analysis of the relationship between slavery and other factors for these countries constitutes the majority fisheries 'signal'. We have edited the manuscript to ensure that this is clear in both the methods and results sections.

Comment: Page 5 - I'm not sure I understand the link between "high levels of unreported catch - low value fisheries – and modern slavery" with respect to the low value bit. Why does low value fisheries go with unreported catch and modern slavery? (Am I simply confused because I'm unclear if 'low value' refers to the total value of the fishery or does it mean low value species?)

Response: Value refers to the mean US\$ value per unit of catch, i.e. a low value fishery (e.g. sardines) generates less revenue per tonne of catch than a high value one (e.g. tuna).

Where fisheries target low value species, there may be an incentive to use forced labour given the sensitivity of the business model to operational costs. We have revised the results section of the manuscript for clarity this point (lines 220-225).

Comment: Page 5 - Japan and Spain shared characteristics with 'high risk' countries. This mention is not further explored but it sparked my interest. It must be something with respect to where they sit on the different dimension in the PCA. It would be good if the authors could explain this a bit more in the discussion.

Response: This is an important point which we failed to bring out in the original draft. An objective of the multivariate PCA analysis was to help us identify countries that might be 'flying under the radar' with respect to modern slavery in fisheries. Both Japan and Spain operate distant water fleets that share characteristics with those of 'problem' countries like South Korea and Taiwan – in particular high levels of subsidies implying that underlying profitability is low. While there have been no media exposés of slavery in their fisheries, Spanish vessel owners operating under flags of convenience have been convicted of serious fisheries crimes (in particular the poaching of Patagonian toothfish) using vessels that may well have been violating labour laws given their modus operandi. Japan operates a 'foreign trainee' programme that has been criticised for abuses and failures to provide adequate safety training and meet other mandatory employer requirements. Neither of these is a smoking gun for labour issues, but we now mention these points in the manuscript to suggest that their similarity to other nations with known labour issues may warrant further investigation (lines 257-266).

Comment: Page 5 – I think when the authors mention low- risk and high-risk they might want to be specific about risk with respect to what. It is easy for the reader to misinterpret risk.

Response: We thank the reviewer for this suggestion, which we have implemented throughout the revised manuscript, and note that it relates to risk of modern slavery.

Comment: Page 6- I'm being nit picky here but when I read that there slavery prevalence is 1.8 victims per 1000 in the EU – I was keen to see the variation in that average (what is the spread). Same for the other averages mentioned in the text.

Response: We have recast this analysis to focus on the lowest risk European countries (see methods and results), but have also included the mean and range as suggest by the reviewer (lines 300-338).

Comment: The authors mention slavery in the context of on-board activities but do not mention illegal 'labour chains' further down the supply chain in the section "Improve supply chain transparency". Is it possible to explore this a bit more deeply? Especially since there might be some gender issues that come into play in that downstream context. This will also mean the already disadvantaged groups are even more disadvantaged.

Response: It is certainly true that labour issues elsewhere in the fisheries supply chain are likely significant, and almost certainly involve more women given their significant role in

processing and sales activities in fisheries in many parts of the world. However, the main focus of our paper is the relationship between the well documented issues of overfishing and the unsustainable management of industrial fisheries, and the relatively new issue of emerging slavery-type employment practices on vessels. The point raised by the reviewer is an important one, and as well as implicitly considering labour issues in all sectors of the economy in our trade analysis, we also acknowledge in the supply chain section of the discussion that improvements to supply chain transparency (and increased vigilance by buyers as to what goes on in their suppliers) will help to uncover and address labour abuses elsewhere in the fisheries value chain (lines 484-490).

Comment: I didn't understand very well how the distant water fishing was dealt with in the analysis. Maybe a better explanation of distant water fleets would be a good start. E.g. On page 1 the authors mention "cost-driven distant-water fleets" but give no reference or explanation why they are cost driven (or more cost driven than other fleets – or subfleets) or why it matters that they are cost driven. By the way – on page 7 the authors refer to costly distant-water fishing operations – which in my mind is something quite different. Something that is not explicitly mentioned in the paper (or might have missed it) is who operates these distant water fishing fleets (which flag countries are they exactly). I think this may need to be explored a bit because it is fundamental to the paper – and the conclusion

Response: We thank the reviewer for this comment as it allows us to improve the clarity of our manuscript. Distant water fishing is any fishing by a country outside its own exclusive economic zone (EEZ), i.e. on the high seas or in the EEZ of any other country. Since these vessels generally travel further and spend longer at sea, and fuel and crew constitute a major share of operating costs and increase with time and distance, these vessels typically incur higher costs than do coastal vessels of the same size and gear type. The large subsidies paid by governments of the major distant water fishing countries, for example as fuel subsidies, supports this observation. Additional evidence in support of this is the tendency of distant water vessel owners to often also own/operate/lease/charter additional distant water support vessels, e.g., refrigerated transshipment vessels (called 'reefers') or refuelling vessels to make the distant water fishing activity possible. Often, such vessels operate under the flags of other states (e.g., flags of convenience) to reduce or avoid licensing and other regulatory costs or to circumvent restrictions on domestic fleet sizes, or to reduce labour and safety-at-sea costs (lines 55-67).

In line with international data reporting mechanisms, all catches are supposed to be reported by the flag state of the vessel (i.e., the flag flown by the fishing vessel) rather than the country of residence of the beneficial owner. The fishing activity modelled in our analysis is therefore that of the flag state reporting the catch on behalf of its flagged fleets. Clearly, flag-hopping, i.e., the tendency by some distant water fleets to regularly and often rapidly re-register to different flags, makes data reporting for distant water fleets even more challenging at times. We have noted this in the methods section.

Comment: "The over-subsidized and poorly supervised distant-water fishing fleets appear to especially foster labour abuses and modern slavery, and the products of these fisheries are consumed in developed countries in significant quantities". If the same nations who import the product also operate the distant water fleets (i.e. the companies that own them can be traced back to Spain or the Netherlands...) then this would seem crucial to the argument. And it might in fact change it a bit.

Response: This is an interesting point, but one that we feel does not impact our assessment. Whilst inconsistencies in flag-state and beneficial ownership will result in the misallocation of fisheries production to countries at the margin, our trade model is based on commodity transactions between nations, not on where particular subsets of fleets land their catches. Thus, if the COMTRADE records show, e.g., Spain to be importing significant quantities of seafood from Thailand, then that seafood can be assumed to have been caught or processed by the Thai industry and so carry a risk of slavery associated with that country. Additionally, not only do the Thai, Taiwanese and Chinese flagged distant water fleets far outnumber foreign-flagged DWF vessels owned by entities based in Europe or North America, the major distant water fishing nations of Asia also operate the lion's share of the 'Flag of Convenience' fleet meaning that allocating catch back to beneficial owners rather than flag states (assuming that that is where catch is landed) will maintain the dominance of these countries, and their labour standards, in the distant-water fishing sector (see methods section on *Slavery and global seafood trade*)

Comment: Page 7 - correlation between structural elements of industrial fisheries and motive and opportunity for unreported fishing and labour abuses. I feel the authors need to explain what those 'structural elements' are that provide that link/correlation. I can 'imagine' them but I feel it needs to be spelt out better as it is crucial for the whole argument of the paper (industrial is mentioned 3 items in the conclusion!).

Response: We thank the reviewer for this comment. We now have elaborated more on the structural elements of contemporary industrial fisheries that seem most likely to contribute to labour abuses – specifically, fleet overcapacity in many regions (driving down per vessel profitability), a dependence on financial subsidies for capital and fuel (suggesting that many fisheries are fundamentally uneconomic), a mode of operation that can readily circumvent meaningful levels of external oversight in all but the wealthiest countries, and, in many of the major fishing countries, a disproportionate reliance on migrant labour (to whom, it seems, domestic norms of treatment need not apply) Lines 341-357.

Comment: I think the discussion has some great suggestions for improving the situation and making slavery less likely to happen. I think that it would be great if the authors could also suggest some research that might help in that regard (I know it is a standard reviewer comment but in this case it would be helpful).

Response: We thank the reviewer for this excellent suggestion. We have included suggestions for avenues for future research in the revised manuscript. (lines 654-665)

Reviewers' comments:

Reviewer #2 (Remarks to the Author):

The authors are to be commended for undertaking some substantial revisions which have greatly improved the quality of the manuscript. I have reviewed the revised manuscript and the response and am satisfied with the revisions, particularly the authors clarification of their approach and their discussion which is much stronger in the depth of recent efforts and potential solutions to address these issues. I feel this paper will be a major contribution to our scientific understanding of these issues, and help provide a wide array of actors with evidence-based pathways forward to reduce social abuses in the seafood sector, together with environmental improvements.

Thank you,
Jack Kittinger

Reviewer #3 (Remarks to the Author):

I appreciate the authors' effort in revising the paper and providing additional explanations in some sections. I would like to reiterate that the modern slavery in fisheries is an important topic, the study is novel, and the subject will be of interest to readers in the many fields. However, I find that no change has been made in the analytical method. Methodological problems and a lack of scientific rigor remain. My review simply focuses on the method, as inadequate analyses have led to unsupported conclusions.

The key objective of the paper is to examine the empirical links between labour abuses (using GSI) and fisheries attributes. The analyses involves two stages. In the first stage, linear models are used to test the overall relationship between slavery prevalence and two fisheries attributes: unreported catch and landed value. In the second stage, a PCA is used to identify the shared characteristics of groups of major fishing countries based on six measures: unreported catch, landed value, catch outside EEZ, GDP, subsidies, and distance of catch. This two-stage approach is a bit strange. Why don't you simply include all these variables in the multiple regression at the first stage as you have done for the first two variables (unreported catch and landed value)? The multiple regression will clearly show what variables are important in predicting slavery. The PCA can reveal what groups are more correlated than others, but does not directly build a linkage between slavery prevalence and these variables as GSI is not used in the analysis. As such, key statements in the abstract and conclusion are not supported by the analyses, for examples:

- Modern slavery and fisheries governance appear closely correlated among the major fishing countries, with the risk of modern slavery highest in countries with weak fisheries enforcement and heavily subsidised distant-water fleets.
- The factors predicting modern slavery mirror those driving overfishing (factors driving overfishing have not been analysed in the paper).
- The concurrent failure by government decision makers, policy developers and fisheries managers in many regions to adapt to the growth and technological changes in industrial fleets has rendered the waters of weaker and poorer countries, and the high seas, open to abuse of both fisheries regulations and international labour standards, thus directly contributing to modern slavery.

Sensitivity analysis presented in the supplementary materials shows a significant impact of uncertainty in the dependent and independent variables on R-square value. Instead of focusing on R-square, measurement errors should be incorporated into the multiple regression models and the data should be re-analysed.

Because you are using the national prevalence estimate of slavery per country, which includes all sectors, what you have done is building a correlation between overall slavery (not slavery in

fisheries) and two selected fisheries attributes. This should be clearly stated in the paper. The current writing may mislead author to a close correlation between slavery in fisheries and the selected two fisheries attributes. Alternatively, a strong assumption is needed that slavery in fishing industry is proportional to overall slavery in the country.

Reviewer #4 (Remarks to the Author):

I have read the authors comments to my review and feel they have addressed them adequately. The concerted effort to address all 4 reviewer comments has improved the paper.

Authors' response to Reviewer #3:

Comment: I appreciate the authors' effort in revising the paper and providing additional explanations in some sections. I would like to reiterate that the modern slavery in fisheries is an important topic, the study is novel, and the subject will be of interest to readers in the many fields. However, I find that no change has been made in the analytical method. Methodological problems and a lack of scientific rigor remain. My review simply focuses on the method, as inadequate analyses have led to unsupported conclusions.

Response: We appreciate that the reviewer sees the value in this timely contribution. However, we are concerned about the continued focus on analytical challenges that cannot be addressed given the acknowledged limitations of existing datasets. In response to the reviewer's original comments, we have (1) completed an uncertainty analysis on the GSI data that shows little influence on the broad patterns; and (2) included clarifications on the modelling approach and provided caveats and qualifications. We have also highlighted in our revised manuscript the importance of the analysis in identifying (1) the characteristics of fisheries that are associated with a higher prevalence of slavery; and (2) countries at particular risk in terms of slavery at sea. The latter can be treated in a sense as a hypothesis to improve data collection in otherwise overlooked fisheries and countries. *The Economist* recently published an article on the well documented issue of forced labour in Thai fisheries – ironically, one of the online comments in response to the article flagged India as free of forced labour in fisheries, a statement that our analysis would contest.

We are encouraged by the fact that our statistical analyses and modelling approach were accepted by the other reviewers, and whilst preferences on approaches may vary, we feel we have provided a clear and sound justification of our approach. To this end, we request an editorial decision that recognises the importance of the topic and our efforts to provide a data-driven framework for the identification of risk factors and potential solutions, within the limitations imposed by the currently available data. With the increased attention this issue deserves to receive, improved and additional data will hopefully be collected over time and will then allow us to refine our approach to deliver higher resolution analyses.

Comment: The key objective of the paper is to examine the empirical links between labour abuses (using GSI) and fisheries attributes. The analyses involve two stages. In the first stage, linear models are used to test the overall relationship between slavery prevalence and two fisheries attributes: unreported catch and landed value. In the second stage, a PCA is used to identify the shared characteristics of groups of major fishing countries based on six measures: unreported catch, landed value, catch outside EEZ, GDP, subsidies, and distance of catch. This two-stage approach is a bit strange. Why don't you simply include all these variables in the multiple regression at the first stage as you have done for the first two variables (unreported catch and landed value)? The multiple regression will clearly show what variables are important in predicting slavery. The PCA can reveal what groups are more correlated than others, but does not directly build a linkage between slavery prevalence and these variables as GSI is not used in the analysis. As such, key statements in the abstract and conclusion are not supported by the analyses.

Response: We deliberately took a two-stage approach in our analyses to lead the reader through our exploration of these data and make apparent the manner in which we had come to our conclusions. The rationale for this approach is provided in the revised manuscript, along with acknowledgements of the caveats associated with trying to perform quantitative analyses in a subject area rife with data collection challenges (line 346-360).

To reiterate our logic, step one of our analytical approach uses a simple regression model to identify two key dimensions of national fisheries that correlate to the prevalence of modern slavery at the national level. This allows us to identify countries that have higher slavery risk, in general, and to test for a hypothesised relationship between the economic (motive) and governance (opportunity) aspects of fisheries activity, and poor human rights records. We are concerned that enacting the reviewer's suggestion that we include all the fisheries variables that are in the PCA into a single multiple regression would result in significant overfitting of the model relative to the two variables used. In specifying the regression model, we used Harrell's one-in-ten rule (Harrell *et al.* 1984, Regression modelling strategies for improved prognostic prediction. *Stat Med.* 3(2): 143–52; Harrell *et al.* 1996, Multivariable prognostic models: issues in developing models, evaluating assumptions and adequacy, and measuring and reducing errors. *Stat Med.* 15(4): 361–87) to limit our model to two predictors, since we only have 20 data points as we focused on major fishing countries. This approach also avoids issues likely to result from significant correlation among some of these variables, which is better accommodated by the principal components model.

Having used linear regression to demonstrate broad concordance between patterns in fisheries performance and slavery prevalence among countries, we then use PCA and clustering processes to differentiate between countries with respect to their risk category for slavery in fisheries specifically. Our approach categorises countries based on both the outcome of the regression and qualitative assessments from the literature, media and seafood industry assessments, taking a multiple lines of evidence approach. Here we are asking the question “what are the attributes that countries in the high, medium and low risk categories share”? The PCA extracts the key information from the broader suite of fisheries and economic attributes hypothesised to influence slavery at sea (for example distant-water fishing away from the oversight of domestic enforcement agencies); k-mean clustering of countries based on their scores on the two most important components identifies the groups. The GSI prevalence data used in the multiple regression are not used at this stage since we acknowledge that they can only measure cross-sector, rather than fisheries-specific slavery risk.

We continue to feel that, given the limits of the data and the fact that this is the first effort to consider broad patterns between risk of slavery and fisheries at a global scale, modelling patterns in slavery among the countries responsible for 80% of the world's industrial landings, such a two-stage approach is appropriate.

Comment: Statement not supported: *“Modern slavery and fisheries governance appear closely correlated among the major fishing countries, with the risk of modern slavery highest in countries with weak fisheries enforcement and heavily subsidised distant-water fleets.”*

Response: We have reworded this sentence in the abstract to clarify the distinction between the conclusions drawn from our two main analyses. Our initial regression analysis finds that the national prevalence of slavery across all sectors, as estimated by the GSI, is correlated with poor fisheries governance as represented by the scale of unreported landings and poor financial performance as represented by the value of catch. The risk of slavery in fisheries specifically, evaluated by triangulating information from the NGO, media and industry reports cited in the manuscript, is then linked in the PCA/clustering analysis to additional factors that differentiate between countries with high, medium and low risk of fostering slavery at sea. We propose that these factors (distant water fishing and high catches outside national waters, high subsidies, and relatively high GDP making countries attractive to migrant labour) act in concert with the economic and governance attributes identified in the linear regression to create circumstances in which labour abuses are both incentivised and made possible. These relationships are discussed in detail in the body of the revised manuscript (lines 361-386).

Comment: Statement not supported: *“The factors predicting modern slavery mirror those driving overfishing (factors driving overfishing have not been analysed in the paper).”*

Response: The synergy between measures to tackle modern slavery at sea and those required to reduce the incentive and opportunity for vessels to overfish, including through illegal fishing, is referenced throughout the section discussing current and proposed mitigation measures. However, we have taken the opportunity in this revision to make equally explicit the fact that the factors driving slavery risk at sea are also those identified as contributing to overfishing (lines 30-35, 382-386, 619-628). Particularly: (1) **subsidies** lead to overcapacity, a race to fish, and **declining per-vessel profitability** and incentivise quota dodging and reductions in crew pay (Sumaila et al. ICES J. Mar. Sci. 65: 832–840.) and (2) **poor governance** and **‘distant-water’ fishing** away from the oversight of domestic enforcement bodies, particularly on the high seas or in the waters of institutionally weaker developing countries, allows such activity impunity. This is reinforced by the use of transshipment by distant water fleets. (Österblom et al. PLoS One 5: e12832; Pauly et al. Fish Fish. 15: 474–488).

Additional aspects of the current fisheries landscape where slavery and overfishing are enabled or caused by common factors are also discussed in our manuscript – namely the need to overcome patchy and inconsistent legal frameworks across jurisdictions (lines 518-542), and lack of supply transparency (lines 544-587).

Comment: Statement not supported: *“The concurrent failure by government decision makers, policy developers and fisheries managers in many regions to adapt to the growth and technological changes in industrial fleets has rendered the waters of weaker and poorer countries, and the high seas, open to abuse of both fisheries regulations and international labour standards, thus directly contributing to modern slavery.”*

Response: We have revised the statement and included suitable references (lines 655-660).

Comment: Sensitivity analysis presented in the supplementary materials shows a significant impact of uncertainty in the dependent and independent variables on R-square value. Instead of focusing on R-square, measurement errors should be incorporated into the multiple regression models and the data should be re-analysed.

Response: Our interpretation of the sensitivity analysis included in the revised manuscript is that it shows no material impact on the overall patterns, notwithstanding expected variation in the coefficient of variation between model simulations. Using the best available global datasets for slavery and fisheries attributes, we see a consistent pattern that is supported by country-specific reports. As discussed above, using multiple linear regression on a larger suite of predictors would overfit our data and is a less preferred analytical strategy. Moreover, the literature on incorporation of measurement errors in regressions largely indicates changes in estimates of slope but no material changes where relationships are strongly significant, as is the case here ($p = 0.006$). We thus prefer to retain our simple and intuitive two-stage approach to modelling slavery in fisheries, and acknowledge the potential impact of sampling error in the sensitivity modelling that was included in the revised manuscript.

Comment: Because you are using the national prevalence estimate of slavery per country, which includes all sectors, what you have done is building a correlation between overall slavery (not slavery in fisheries) and two selected fisheries attributes. This should be clearly stated in the paper. The current writing may mislead reader to a close correlation between slavery in fisheries and the selected two fisheries attributes. Alternatively, a strong assumption is needed that slavery in fishing industry is proportional to overall slavery in the country.

Response: The distinction requested by the reviewer is already clear in the manuscript but we have attempted to further clarify the matter, reiterating at several points that our initial regression analysis relies on national cross-sectoral estimates as these are the only ones currently available (e.g. lines 199-201, 207-212, 237-239). We are also clear that, while countries which allow slavery to flourish in, for instance, the garment sector are likely challenged in fisheries as well, the GSI is currently insufficiently precise a tool to allow detailed conclusions about country-specific risks in slavery at sea (lines 239-242, 346-357). It is for this reason that we triangulate across other information sources to drill deeper into the risk factors for fisheries slavery in the analyses which follow our exploratory regression modelling. As we acknowledge in the closing part of the discussion, additional work focussed specifically on quantifying slavery on fishing vessels is required to improve on this first analysis (lines 677 onwards). However, we feel that, data limitations notwithstanding,

our conclusions are supported by multiple lines of evidence from the investigations completed to date.

REVIEWERS' COMMENTS:

Reviewer #3 (Remarks to the Author):

I appreciate the author's response. However, my concerns on methods section and their interpretations of the results remain. I focus on the analytical methods because they provide the information by which the validity of a study is ultimately judged. If analytical challenges cannot be addressed given the acknowledged limitations of existing datasets, the paper should focus on what can be done and not over-interpret the results.

I have no problem with the conclusion "a correlation between increased prevalence of country-level modern slavery and higher levels of unreported catches and lower mean value of the catch of industrial fisheries for the 20 countries" (Lines 658-660). However, this conclusion has been overly extended to other factors, such as the statement in the abstract "the risk of labour abuses at sea is highest in countries with weak fisheries enforcement and heavily subsidised distant water fleets". If subsidies and distance of catch are deemed to be important, these two predictors should replace the mean value and be used in the linear regression model.

The author argued that including all the fisheries variables that were in the PCA into a single multiple regression would result in significant overfitting of the model relative to the two variables used. It seems to me that a model with 6 predictors could be tried using 20 data points. Furthermore, there are two alternatives: (i) focus on the three predictors that appear to be most interesting: unreported catches, subsidies, and distance of catch; (2) use all countries instead of arbitrarily selecting the top 20.

I agree with the authors' request that the Editor should make the decision based on all information received.

Reviewer #4 (Remarks to the Author):

Comment provided to editor

Response to Reviewer comments:

Reviewer #3

Comment: I appreciate the author's response. However, my concerns on methods section and their interpretations of the results remain. I focus on the analytical methods because they provide the information by which the validity of a study is ultimately judged. If analytical challenges cannot be addressed given the acknowledged limitations of existing datasets, the paper should focus on what can be done and not over-interpret the results.

Response: We acknowledge the limitations of current data in the manuscript (see response to Editor above), and have edited throughout to ensure that we do not over-interpret what are, by necessity, preliminary results in analysing the slavery-fisheries space. Our intention is to identify apparent correlations and potential drivers within the available data on slavery and fisheries, and the hypotheses suggested by them. In doing this we offer a framework within which subsequent analysis may be contextualised and prioritised.

Comment: I have no problem with the conclusion "a correlation between increased prevalence of country-level modern slavery and higher levels of unreported catches and lower mean value of the catch of industrial fisheries for the 20 countries" (Lines 658-660). However, this conclusion has been overly extended to other factors, such as the statement in the abstract "the risk of labour abuses at sea is highest in countries with weak fisheries enforcement and heavily subsidised distant water fleets". If subsidies and distance of catch are deemed to be important, these two predictors should replace the mean value and be used in the linear regression model.

Response: In revising the manuscript to address the comments of the Reviewer and the Editor, we have been careful to address the limitations of our analyses. We have also made clear the distinction between the first two analyses we present. The initial regression analysis uses the GSI slavery measure as a proxy for potential slavery risk in fisheries to test for a correlation between national-level slavery-type practices and poor fisheries governance at the national level. The subsequent PCA uses a range of sources to qualitatively identify countries as high, medium or low risk for fisheries slavery, based on known and reported labour issues. Hence, we do not rely on the regression analysis to make inferences about the role of distant-water fleets in cases of slavery. Rather, we report that the PCA and clustering approach find that the majority of countries with documented serious labour issues have a high proportion of fishing at a great distance from 'home' ports and harmful state subsidies known to foster overcapacity and support unprofitable fisheries.

Comment: The author argued that including all the fisheries variables that were in the PCA into a single multiple regression would result in significant overfitting of the model relative to the two variables used. It seems to me that a model with 6 predictors could be tried using 20 data points. Furthermore, there are two alternatives: (i) focus on the three predictors that appear to be most interesting: unreported catches, subsidies, and distance of catch; (2) use all countries instead of arbitrarily selecting the top 20.

Response: Firstly, there is a general statistical rule of thumb that you need at least 10 observations per independent variable in a regression, with some authors suggesting 20 or more for stepwise regression. To that end, six predictors in a model with 20 samples would result in overfitting with consequences for model coefficients and significance. Secondly, the regression and PCA analyses are based on different sets of dependent (slavery) variables. The regression correlates the quantitative GSI country-level measure as a proxy, recognising that it may under- or overestimate slavery at sea in some countries and does so as a function of national fisheries characteristics. The PCA, on the other hand, seeks to identify common fisheries-related drivers that group countries together based on a qualitative measure of fisheries slavery risk derived from reported slavery cases, i.e., the GSI is not used in the PCA. Finally, the choice of using the top 20 countries for both analyses is based on their overwhelming contribution to wild-caught seafood – 80% of global industrial supply – rather than some arbitrary selection criteria. Any work targeting a reduction in slavery in fisheries would likely prioritise assessment of these countries ahead of smaller contributors to seafood supply, that each contribute only a fraction of the remaining 20% of global supply.

Comment: I agree with the authors' request that the Editor should make the decision based on all information received.

Response: The Editor has provided extensive comments and direction in addition to the Reviewer's comments above. We have incorporated responses to both into the revised manuscript.